# CONTROLLABLE IMAGE GENERATION VIA COLLAGE REPRESENTATIONS

## ABSTRACT

Recent advances in conditional generative image models have enabled impressive results. On the one hand, text-based conditional models have achieved remarkable generation quality, by leveraging large-scale datasets of image-text pairs. To enable *fine-grained* controllability, however, text-based models require long prompts, whose details may be ignored by the model. On the other hand, layout-based conditional models have also witnessed significant advances. These models rely on bounding boxes or segmentation maps for precise spatial conditioning in combination with coarse semantic labels. The semantic labels, however, cannot be used to express detailed appearance characteristics. In this paper, we approach fine-grained scene controllability through image collages which allow a rich visual description of the desired scene as well as the appearance and location of the objects therein, without the need of class nor attribute labels. We introduce "mixing and matching scenes" (M&Ms), an approach that consists of an adversarially trained generative image model which is conditioned on appearance features and spatial positions of the different elements in a collage, and integrates these into a coherent image. We train our model on the OpenImages (OI) dataset and evaluate it on collages derived from OI and MS-COCO datasets. Our experiments on the OI dataset show that M&Ms outperforms baselines in terms of fine-grained scene controllability while being very competitive in terms of image quality and sample diversity. On the MS-COCO dataset, we highlight the generalization ability of our model by outperforming DALL-E in terms of the zero-shot FID metric, despite using two magnitudes fewer parameters and data. Collage based generative models have the potential to advance content creation in an efficient and effective way as they are intuitive to use and yield high quality generations.

## 1 INTRODUCTION

Controllable image generation leverages user inputs – *e.g.* textual descriptions, scene graphs, bounding box layouts, or segmentation masks – to guide the creative process of composing novel scenes. Text-based conditionings offer an intuitive mechanism to control content creation, and short and potentially high level descriptions can result in high quality generations (Ding et al., 2021; Gafni et al., 2022; Nichol et al., 2022; Ramesh et al., 2021; Reed et al., 2016; Rombach et al., 2022). However, to describe complex scenes in detail, long text prompts are required, which are challenging for current models, see *e.g.* the person's position in the second row of Figure 1. Moreover, current text-based models require very large training datasets composed of tens of millions of data points to obtain satisfactory performance levels. Bounding boxes (BB) (Sun & Wu, 2019; 2020; Sylvain et al., 2021; Zhao et al., 2019), scene graphs (Ashual & Wolf, 2019) and segmentation mask (Chen & Koltun, 2017; Liu et al., 2019; Park et al., 2019; Qi et al., 2018; Schönfeld et al., 2021; Tang et al., 2020b; Wang et al., 2018; 2021) conditionings offer strong spatial and class-level semantic control, but offer no control over the appearance of scene elements beyond the class level. Although user interaction is still rather intuitive, the diversity of the generated scenes is often limited, see *e.g.* the third and fourth rows of Figure 1, and the annotations required to train the models are laborious to obtain. Moreover, the generalization ability of these approaches is restricted by the classes and scene compositions appearing in the training set (Casanova et al., 2020).

In this paper, we explore fine-grained scene generation controllability by leveraging image collages to condition the model. As the saying goes, *a picture is worth a thousand words*, and therefore,

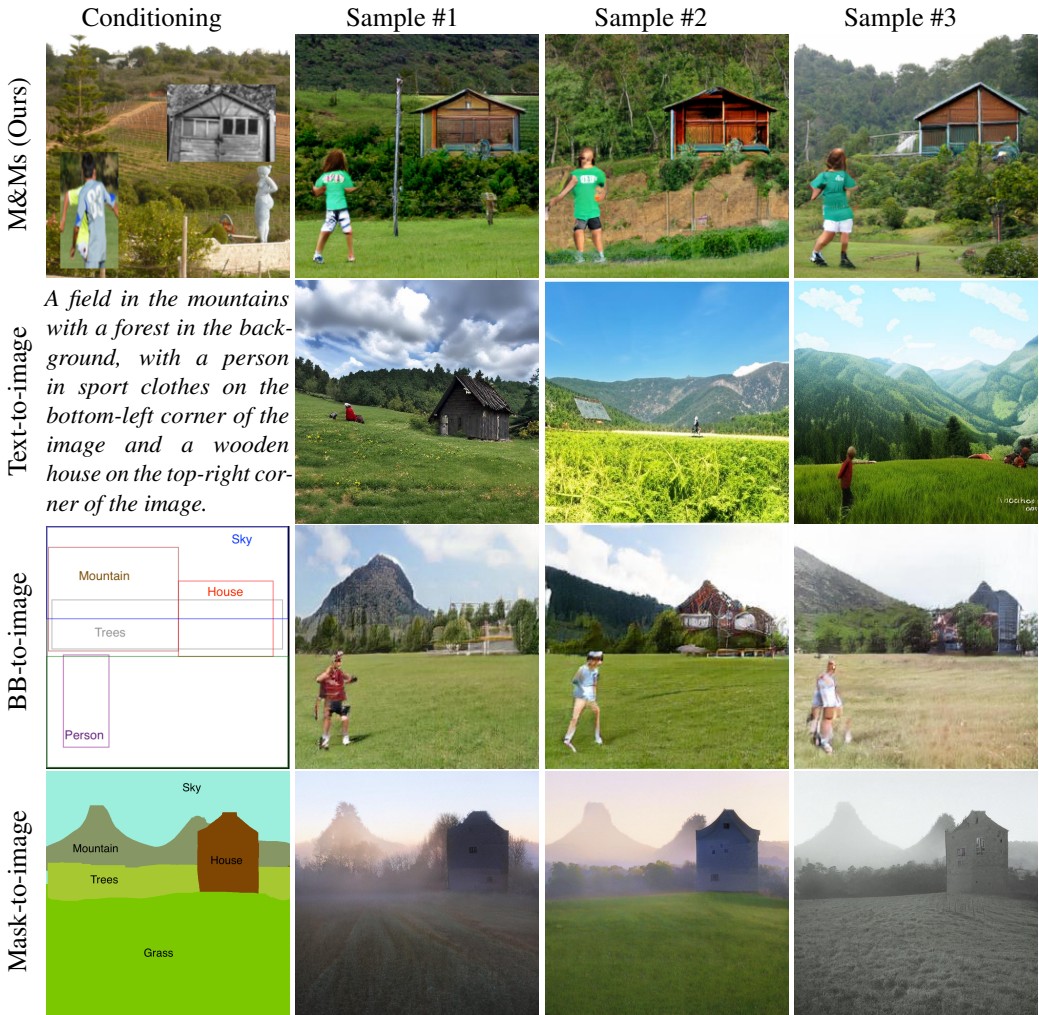

Figure 1: Approaches to controllable scene generation (from top): our approach based on collages (M&Ms), text-to-image model (Make-a-scene (Gafni et al., 2022), samples are a courtesy of the paper authors), BB-to-image model (LostGANv2 (Sun & Wu, 2020)), and Mask-to-image (GauGAN2 (Park et al., 2019)). Note that the models have been trained on different datasets and, as such, GauGAN2 cannot generate people as the model was not trained on a dataset containing this class. Visualization of the input collage to M&Ms is depicted as a collaged RGB image for simplicity.

the rich information contained in image collages has the potential to effectively guide the scene generation process, see the first row of Figure 1. Collage-based conditionings can be easily created from a set of images with minimal user interaction, and provide a detailed visual description of the scene appearance and composition. Moreover, as image collages by construction do not require any semantic labels, leveraging them holds the promise of benefiting from very large and easy-to-obtain datasets to improve the resulting image quality.

To enable collage-based scene controllability, we introduce Mixing & Matching scenes (M&Ms), an approach that extends the instance conditioned GAN (IC-GAN, (Casanova et al., 2021)) by leveraging image collages and treating each element of the collage as a separate instance. In particular, M&Ms takes as input an image collage, extracts representations of each one of its elements, and spatially arranges these representations to generate high quality images that are similar to the input collage. M&Ms is composed of a pre-trained feature extractor, a generator and two discriminators, operating at image and object level respectively. Similar to IC-GAN, M&Ms leverages the neighbors of the instances in the feature space to model local densities but extends the framework to blend multiple localized distributions – one per collage element – into coherent images. We train M&Ms on the OpenImages dataset (Kuznetsova et al., 2020) and validate it using collages derived from OpenImages

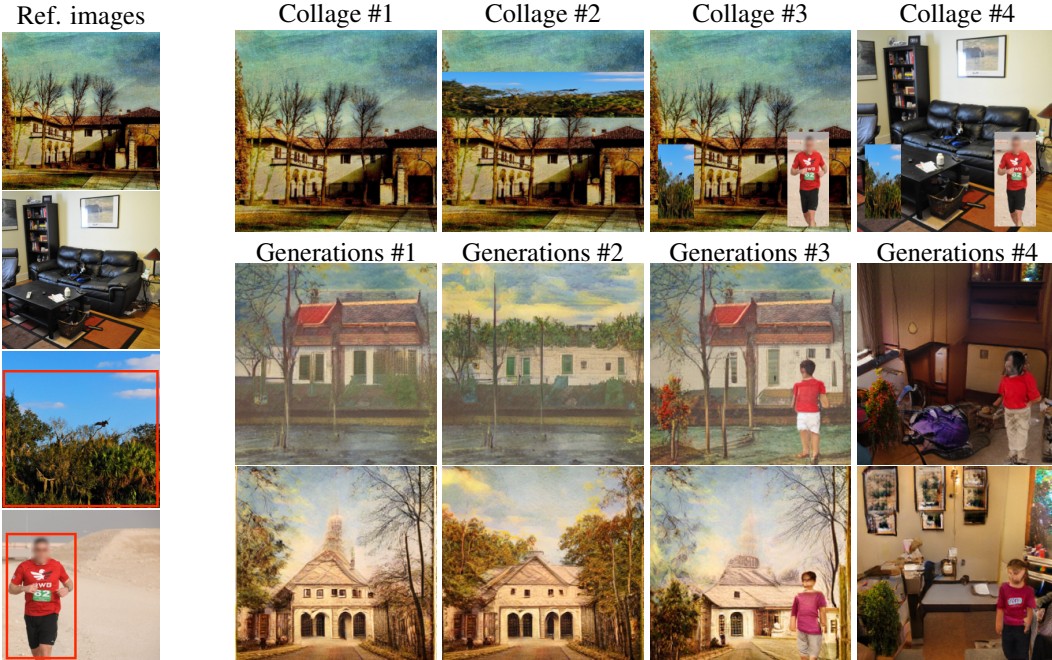

Figure 2: Images generated with M&Ms from four different collages. We obtained collages from four reference images (leftmost column) coming from OpenImages and MS-COCO. Collages #1, #2 and #3 are obtained by combining only the OpenImages images, while the collage #4 highlights the zero-shot performance of M&Ms as it contains elements both from OpenImages and MS-COCO images. Collage #1 consists of a background scene without added objects. Note that input collages to the model are depicted as RGB overlays. Faces in the input images have been blurred out.

images. We further challenge the model by conditioning it on collages composed of MS-COCO (Lin et al., 2014) images, not seen during training. We show that M&Ms is able to generate appealing and diverse images in all cases, exhibiting zero-shot generalization to collages composed of scenes, objects and layouts unseen during training. Figure 2 presents M&Ms in action. First, we select four images, three from OpenImages and one from MS-COCO, to compose our collages and we draw BBs to crop the objects of interest (left most column). Next, we create four collages by combining each scene with either zero (second column), one (third column) or two (fourth and fifth columns) cropped objects, adjusting their BBs, and use M&Ms to obtain two samples for each resulting collage. Note that the fifth row combines a background from MS-COCO and objects from OpenImages. We observe that for each created collage, M&Ms generates diverse scenes that are consistent with the collage layout, captures the style of each element, and blends all the collage elements into coherent images, see generations #1 – #4.

Our contributions can be summarized as follows: (1) We introduce M&Ms, a model to explore collage based conditioning as a simple yet effective way to control scene generation; (2) On OpenImages, we show that M&Ms generates coherent images even from collages that combine random objects and scenes, and offers superior scene controllability w.r.t. baselines without affecting the generated image quality; (3) To showcase the zero-shot transfer of our model, we evaluate it on the MS-COCO dataset showing that M&Ms surpasses DALL-E (Ramesh et al., 2021) in terms of the zero-shot FID metric despite being trained on a two orders of magnitude smaller dataset, and having two orders of magnitude less parameters.

## 2  CONTROLLABLE IMAGE GENERATION VIA COLLAGES

In this section, we briefly review IC-GAN in Section 2.1 and present its extension to handle collages through Mixing & Matching scenes (M&Ms) in Section 2.2.

## 2.1 REVIEW OF INSTANCE-CONDITIONED GAN

Instance-Conditioned GAN (IC-GAN) (Casanova et al., 2021) considers the data manifold as a set of fine-grained and overlapping clusters, where each cluster is defined by a datapoint – or *instance* – $\mathbf{x}_i$ and the set of its $k$ nearest neighbors $\mathcal{A}_i$ in a given feature space. IC-GAN models the data distribution $p(\mathbf{x})$ as a mixture of instance-conditioned distributions, such that $p(\mathbf{x}) \approx \frac{1}{M} \sum_i p(\mathbf{x}|\mathbf{h}_i)$, where $M$ is the number of datapoints in a given dataset and $\mathbf{h}_i = f_\theta(\mathbf{x}_i)$ are the instance features of $\mathbf{x}_i$ obtained with an embedding function $f$ parameterized by $\theta$. The conditional distributions $p(\mathbf{x}|\mathbf{h}_i)$ are modelled by a generator $G$ whose inputs are a noise vector $\mathbf{z} \sim \mathcal{N}(0, I)$ sampled from a unit Gaussian, and instance features $\mathbf{h}_i$, such that the generated image can be obtained as $G(\mathbf{z}, \mathbf{h}_i)$. The discriminator is conditioned on the same instance features $\mathbf{h}_i$ and tries to distinguish between the generated samples and the real images $\mathbf{x}_j$ neighboring $\mathbf{x}_i$, which are sampled from $\mathcal{A}_i$. Both generator and discriminator engage in a two-player min-max game to find the Nash equilibrium of the following equation:

$$\min_G \max_D \mathbb{E}_{\mathbf{x}_i \sim p(\mathbf{x}), \mathbf{x}_j \sim \mathcal{A}_i} \left[ \ln D(\mathbf{x}_j, \mathbf{h}_i) \right] + \mathbb{E}_{\mathbf{x}_i \sim p(\mathbf{x}), \mathbf{z} \sim \mathcal{N}(0, I)} \left[ \ln(1 - D(G(\mathbf{z}, \mathbf{h}_i), \mathbf{h}_i)) \right], \quad (1)$$

where we use $\mathbf{x}_j \sim \mathcal{A}_i$ to denote that $\mathbf{x}_j$ is sampled uniformly from the set $\mathcal{A}_i$.

Despite its ability to generate visually compelling samples, IC-GAN can only control the generated images through a single conditioning image (and a class label in its class-conditional version).

## 2.2 MIXING & MATCHING SCENES (M&MS)

We define an image collage as a list $C = [\mathbf{s}, \mathcal{O}]$ containing one *background scene image*, $\mathbf{s}$, and a set of *foreground collage elements*, $\mathcal{O}$. The set of foreground collage elements is defined as a set of tuples $\mathcal{O} = \{(\mathbf{o}_c, \mathbf{b}_c)\}_{c=1}^O$, where $\mathbf{o}_c$ is a foreground object image and $\mathbf{b}_c$ its corresponding bounding box, indicating the location and size of $\mathbf{o}_c$ in the collage. Consider now a dataset $\mathcal{D} = \{(C_i, \mathbf{x}_i)\}_{i=1}^M$ composed of collages $C_i$ and corresponding natural images $\mathbf{x}_i$. The objective of collage-based image generation is to model $p(\mathbf{x}|C_i)$.

In M&Ms, we model $p(\mathbf{x}|C_i)$ as $p(\mathbf{x}|\mathbf{H}_i)$, where $\mathbf{H}_i$ is a representation tensor of the image collage $C_i$. We use a pre-trained neural network $f$ parameterized by $\phi$ to obtain a representation of the background scene image $\mathbf{h}_i^s = f_\phi(\mathbf{s}_i)$ and of each collage element $\mathbf{h}_{i,c}^o = f_\phi(\mathbf{o}_{i,c})$. We then spatially replicate $\mathbf{h}_i^s$ and all $\mathbf{h}_{i,c}^o$ into $\mathbf{H}_i$ using the collage bounding boxes $\mathbf{b}_{i,c}$, as illustrated in Figure 3a. When two bounding boxes overlap, we assume that the smaller box is in front of the bigger one.

We train M&Ms adversarially using one generator and two discriminators: a global discriminator and an object discriminator. The objective of the global discriminator is to encourage the generator to produce plausible and appealing images, while the object discriminator ensures that the generated objects are both of high visual quality and consistent with the collage. Our generator $G(\mathbf{Z}_i, \mathbf{H}_i)$ takes as an input a noise tensor $\mathbf{Z}_i$ and the representation tensor of the collage $\mathbf{H}_i$ and outputs a generation $\hat{\mathbf{x}}$. The noise tensor is composed of a scene noise vector $\mathbf{z}^s \sim \mathcal{N}(0, \mathbf{I})$ and $O$ object noise vectors $\mathbf{z}_c^o \sim \mathcal{N}(0, \mathbf{I})$ that are spatially arranged into $\mathbf{Z}_i$ using the collage bounding boxes $\mathbf{b}_{i,c}$. The generation process is shown in Figure 3b.

The design of the training procedure for both discriminators follows the spirit of IC-GAN – *i.e.* during training we ensure that the generated images and its objects are within the neighbourhood of $\mathbf{x}_i$ and $\mathbf{o}_{i,c}$, respectively. We use the dataset $\mathcal{D}$ to obtain $k$ nearest neighbour sets $\mathcal{A}_i^x$ and $\mathcal{A}_{i,c}^o$ for images $\mathbf{x}_i$ and object images $\mathbf{o}_{i,c}$, respectively. The neighbours are computed using the image and object representations $\mathbf{h}_i^x$ and $\mathbf{h}_{i,c}^o$ by means of the cosine similarity, where $\mathbf{h}_i^x = f_\phi(\mathbf{x}_i)$. As depicted in Figure 3b, the global discriminator $D^g(\mathbf{x}, \mathbf{h}_i^x)$ discerns between real neighbors sampled from $\mathcal{A}_i^x$ and images generated by conditioning on $\mathbf{H}_i$, while the object discriminator $D^o(\mathbf{o}, \mathbf{h}_{i,c}^o)$ discerns between real neighbors sampled from $\mathcal{A}_{i,c}^o$ and objects in the generated image, which are extracted using the bounding boxes $\mathbf{b}_{i,c}$. Note that M&Ms reverts to IC-GAN if the image collage only contains a single background scene image.

**Training M&Ms.** The generator and discriminators attempt to find the Nash equilibrium of the following equation:

$$\min_G \max_{\{D^g, D^o\}} (1 - \lambda)\mathcal{L}^g(G, D^g) + \lambda\mathcal{L}^o(G, D^o), \quad (2)$$

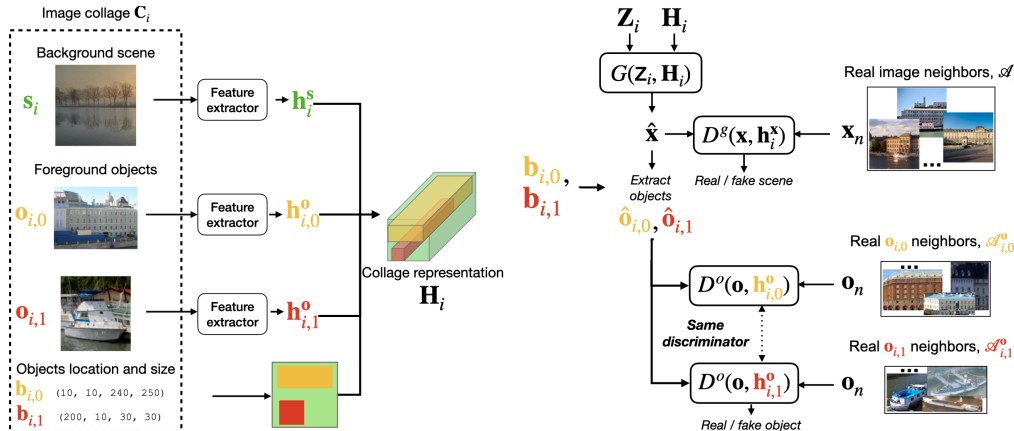

(a) Process of extracting a collage representation $\mathbf{H}_i$.      (b) Training workflow of M&Ms.

Figure 3: Overview of M&Ms. (a) Example collage $C_i$ with one background scene image, $\mathbf{s}_i$, and two foreground objects, $\mathbf{o}_{i,0}$ and $\mathbf{o}_{i,1}$. We extract feature representations, $\mathbf{h}_i^s$, $\mathbf{h}_{i,0}^o$, and $\mathbf{h}_{i,1}^o$, from the background scene image and the foreground object images, respectively. The object locations and sizes in the collage, $\mathbf{b}_{i,0}$ and $\mathbf{b}_{i,1}$, are used in conjunction with the extracted feature representations to spatially build the collage representation $\mathbf{H}_i$. (b) We feed the generator $G$ with the extracted collage representation $\mathbf{H}_i$ and a noise tensor $\mathbf{Z}_i$, and generate an image $\hat{\mathbf{x}}$. The global discriminator $D^g$ distinguishes between generated images and real image neighbors, while the object discriminator $D^o$ differentiates between generated objects and real object neighbors. Some weights are shared for the global and object discriminator (see Section C.2 in Supplementary Material for details).

where $\mathcal{L}^g$ is the global loss, $\mathcal{L}^o$ is the object loss, and $\lambda$ controls the importance of each term. The global and object losses are defined as:

$$\mathcal{L}^g = \mathbb{E}_{(C_i, \mathbf{x}_i) \sim \mathcal{D}, \mathbf{x}_n \sim \mathcal{A}_i^x} \left[ \log D^g(\mathbf{x}_n, \mathbf{h}_i^x) \right] + \mathbb{E}_{(C_i, \mathbf{x}_i) \sim \mathcal{D}, \mathbf{Z}_i \sim \mathcal{N}(0, \mathbf{I})} \left[ \log(1 - D^g(\hat{\mathbf{x}}, \mathbf{h}_i^x)) \right]. \quad (3)$$

$$\begin{aligned} \mathcal{L}^o &= \mathbb{E}_{(C_i, \mathbf{x}_i) \sim \mathcal{D}, (\mathbf{o}_{i,c}, \mathbf{b}_{i,c}) \sim \mathcal{C}_i, \mathbf{o}_n \sim \mathcal{A}_{i,c}^o} \left[ \log D^o(\mathbf{o}_n, \mathbf{h}_{i,c}^o) \right] \\ &+ \mathbb{E}_{(C_i, \mathbf{x}_i) \sim \mathcal{D}, (\mathbf{o}_{i,c}, \mathbf{b}_{i,c}) \sim \mathcal{C}_i, \mathbf{Z}_i \sim \mathcal{N}(0, \mathbf{I})} \left[ \log(1 - D^o(\hat{\mathbf{o}}_{i,c}, \mathbf{h}_{i,c}^o)) \right], \quad (4) \end{aligned}$$

where $\hat{\mathbf{o}}_{i,c}$ corresponds to the generated object extracted from the generated image $\hat{\mathbf{x}}$ using $\mathbf{b}_{i,c}$, and $\mathbf{Z}_i \sim \mathcal{N}(0, \mathbf{I})$ refers to the sampling procedure of the noise tensor.

In practice, obtaining the ground truth collage decompositions of natural images is challenging. However, approximate decompositions can be easily obtained from the $\mathbf{x}_i$, by applying a set of class-agnostic bounding boxes to extract object images and assuming that the background scene image $\mathbf{s}_i$ is equivalent to $\mathbf{x}_i$. In the absence of class agnostic bounding box annotations, bounding boxes can be obtained by leveraging an object proposal model such as OLN (Kim et al., 2022), as validated in Supplementary Material G.

**M&Ms architecture details.** We start from the BigGAN generator and discriminator architectures (Brock et al., 2019) given their ubiquitous and successful use in the literature, and extend them to handle collages. In particular, we modify the normalization layers in the generator to take as input a collage representation $\mathbf{H}_i$ (instead of a class embedding) and replace the fully connected layers by 1×1 convolutions. We also adapt the discriminator architecture to operate both at image level and object level, by stacking ResNet blocks which are subsequently fed through an image processing path and an object processing path, tying the parameters of the M&Ms's image and object discriminators. The image processing path applies a global pooling after the last ResNet block. The object processing path applies a region of interest (RoI) pooling layer (Ren et al., 2015) at four intermediate ResNet block outputs. We then compute the dot product between the output of the image processing path and the real image features $\mathbf{h}_i^x$ to obtain the image discriminator score. Analogously, we compute the dot product between the outputs of the object processing path and the features of the real objects $\mathbf{h}_{i,c}^o$ to obtain the object discriminator scores. Note that this follows the projection discriminator (Miyato & Koyama, 2018) principle of BigGAN, except that we project on image and object features rather than class embeddings. More details can be found in the Supplementary Material C.2.

## 3 EXPERIMENTS

Here, we first describe the datasets, evaluation metrics and training details. Then, we present quantitative and qualitative results when training on OpenImages and evaluating on the same dataset. Additionally, we transfer M&Ms to MS-COCO and demonstrate zero-shot generalization.

### 3.1 EXPERIMENTAL SETUP

**Dataset.** We train M&Ms on the 1.7M images of the OpenImages (v4) dataset (Kuznetsova et al., 2020) and extract collages using the ground-truth bounding box annotations, see training details of Section 2.2. We validate our model in both in-distribution and out-of-distribution scenarios using collages derived from OpenImages images as well as in a zero-shot dataset scenario using collages derived from MS-COCO (Lin et al., 2014) images.

**Metrics.** We measure the quality and diversity of the generated images by means of the Fréchet Inception Distance (FID) (Heusel et al., 2017). More precisely, we measure the image FID ($FID_x$) on generated images and the object FID ($FID_o$) on generated objects. We obtain the reference distributions to compute $FID_x$ and $FID_o$ using the training set of OpenImages or the validation set of MS-COCO. We obtain the generated data distribution by sampling 50k samples from the model, conditioning on a randomly sampled collage each time. Metrics are reported as the mean and standard deviation over five random seeds that control the randomness of conditionings and noise vectors.

Training details are described in Supplementary Material C.3. Additionally, sensitivity studies regarding the choice of the neighborhood sizes, the object/scene trade-off controlled by $\lambda$, and the number of objects per scene, as well as ablations wrt the usage of object features versus class labels can be found in Supplementary Material F.

### 3.2 RESULTS

Here, we present three experiments: (1) we evaluate how M&Ms fits the training data on OpenImages v4; (2) we create random collages from OpenImages images to test the controllability enabled by our model; (3) we challenge M&Ms with zero-shot transfer by evaluating the model on collages from MS-COCO. Additional experiments can be found in Supplementary Materials D and G, including fidelity metrics, comparison to additional baselines and training our model without labeled data.

**Fitting the training data.** In this case, the collage representations of M&Ms are based on scene and object features extracted directly from the training images and their bounding boxes.

In Table 1, we compare M&Ms to IC-GAN, which is conditioned on the global scene background features only. M&Ms achieves significantly better (lower) object FID than IC-GAN, while maintaining roughly the same image FID, highlighting the benefit of leveraging collage representations. To zoom in into the image level results, we compute precision ($P_x$) and recall ($R_x$) metrics (Kynkaan-niemi et al., 2019) at image level: M&Ms results in lower precision than IC-GAN, while exhibiting higher recall. This is perhaps unsurprising, given the M&Ms training. In particular, the M&Ms discriminator considers both objects and scene neighbours as real samples, therefore potentially

Table 1: Quantitative results on $256 \times 256$ OpenImages. *: reported as 0.0 due to rounding.

|  | IC-GAN | M&Ms (ours) |
|---|---|---|
| ↓$FID_x$ | **5.9** ± 0.1 | 6.1 ± 0.1 |
| ↓$FID_o$ | 23.8 ± 0.1 | **8.3** ± 0.1 |
| ↑$P_x$ | **69.6** ± 0.4 | 67.3 ± 0.5 |
| ↑$R_x$ | 67.6 ± 0.3 | **69.4** ± 0.4 |

pushing the overall image generations further away from the training data manifold. Intuitively, when using the same neighbourhood size at scene level, M&Ms can go beyond what is captured within this neighbourhood by introducing changes coming from the objects and their neighbours. This results in a drop in precision. At the same time, the increase in recall reflects a higher coverage as the real data lies within the generated data manifold more often.

**Out-of-distribution collages.** To assess the robustness of M&Ms to out-of-distribution conditionings, we build five types of random collages not seen during training. We start from a real image and decompose it into a collage. We then replace up to $O$ objects in the collage by altering either the object images or their bounding boxes. We alter object images by replacing them by another object

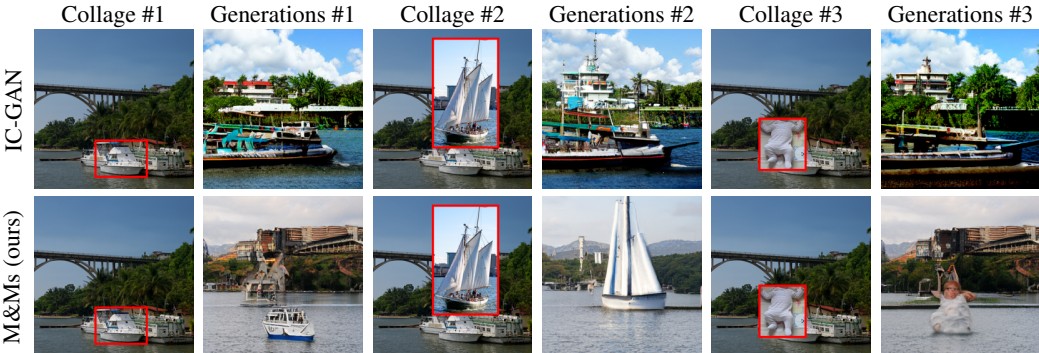

Figure 4: Example of collage conditionings and generated samples of IC-GAN and M&Ms, both trained on OpenImages (256×256). We depict three scenarios: (Collage #1) a collage with one boat object; (Collage #2) a collage where the original object has been replaced with a random object of the same class; (Collage #3) a collage where the original object has been replaced with a random dataset object. In all cases, the bounding box of the randomly sampled object is used. Note that input collages to the M&Ms model are depicted as RGB overlays with red boxes to ease visualization. For IC-GAN, the RGB overlay is used as the input image, and no bounding box information is used.

image sampled uniformly from (1) the original object's neighbors ($S_N$); (2) the dataset objects of the same class ($S_C$); or (3) the entire dataset of objects ($S_R$). We either keep the original object's bounding box ($S_O$) or use the bounding box corresponding to the sampled object ($S_S$). We compare M&Ms with IC-GAN in terms of object and image FID, following the above-mentioned out-of-distribution scenarios. For IC-GAN, we convert the collages into RGB images by blending the object images with the background scene. Results are reported in Table 2.

When comparing the models, we observe that M&Ms consistently outperforms IC-GAN in terms of image FID and, by a large margin, in terms of object FID as well. This suggests that the collage representation used to condition M&Ms is better suited than the one of IC-GAN to enhance generation controllability. Moreover, the object FID results of M&Ms suggest that the object quality and diversity is preserved even in the most out-of-distribution scenario ($S_R$, $S_S$), which considers random objects and bounding box changes. In terms of image FID, the challenging scenarios with random objects ($S_R$, $S_S$) and ($S_R$, $S_O$) obtain the highest scores. We hypothesize that this is in part due to the enabled controllability, which results in generations that are far away from the reference images, changing the distribution of generated images and resulting in higher image FID. As illustrated in Figure 4, and in contrast to IC-GAN, M&Ms results in generations that respect the object appearance and size of the conditioning collage, highlighting the generation controllability of our method. More qualitative results can be found in Supplementary Material E.

Table 2: Quantitative results on OpenImages (256×256) when using random collages obtained by altering either the object images or their bounding boxes (BB). Objects can be replaced by another object sampled from: the original object's neighbourhood ($S_N$), the original object's class ($S_C$), or the entire dataset of objects ($S_R$). We either keep the BB corresponding to the original object ($S_O$) or use the BB corresponding to the sampled object ($S_S$). *: reported as 0.0 due to rounding.

|  | Object | BB | ↓$FID_x$ | ↓$FID_o$ |
|---|---|---|---|---|
| **IC-GAN** | $S_N$ | $S_O$ | $7.3 \pm 0.1$ | $24.5 \pm 0.2$ |
|  | $S_C$ | $S_O$ | $14.9 \pm 0.2$ | $30.7 \pm 0.3$ |
|  | $S_R$ | $S_O$ | $18.8 \pm 0.1$ | $33.0 \pm 0.3$ |
|  | $S_N$ | $S_S$ | $6.8 \pm 0.0$ | $22.6 \pm 0.1$ |
|  | $S_C$ | $S_S$ | $11.1 \pm 0.1$ | $28.1 \pm 0.2$ |
|  | $S_R$ | $S_S$ | $16.6 \pm 2.8$ | $30.9 \pm 2.9$ |
| **M&Ms (ours)** | $S_N$ | $S_O$ | $5.9 \pm 0.0^*$ | $8.2 \pm 0.1$ |
|  | $S_C$ | $S_O$ | $9.1 \pm 0.0^*$ | $8.8 \pm 0.1$ |
|  | $S_R$ | $S_O$ | $15.2 \pm 0.1$ | $11.4 \pm 0.1$ |
|  | $S_N$ | $S_S$ | $5.9 \pm 0.0^*$ | $9.8 \pm 0.1$ |
|  | $S_C$ | $S_S$ | $9.6 \pm 0.0^*$ | $8.3 \pm 0.1$ |
|  | $S_R$ | $S_S$ | $16.0 \pm 0.1$ | $9.2 \pm 0.1$ |

**Zero-shot transfer to MS-COCO.** To demonstrate we can go beyond the collages derived from the training data, we condition M&Ms on collages composed of MS-COCO (Lin et al., 2014) validation images. More precisely, we decompose each image in the dataset into a collage using the ground truth bounding box annotations, and use the resulting collages to condition a pre-trained M&Ms. Table 3 compares M&Ms with recent text-to-image models in terms of zero shot FID. This comparison is meant as a gauge of how M&Ms fares in this setup, rather than a direct comparison with text-to-image models. M&Ms achieves competitive zero-shot FID, surpassing DALL-E, despite using a

| Collages | Generations #1 | Generations #2 | Generations #3 |

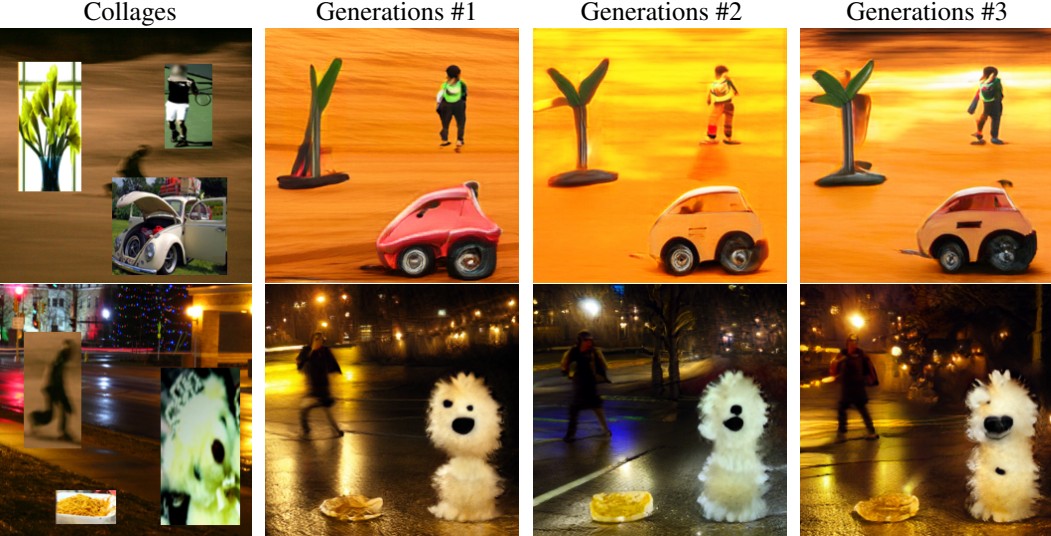

Figure 5: User-generated collages derived from the MS-COCO dataset, and the corresponding M&Ms generated images. The object combinations and layouts are unseen with respect to the training data. Input collages to the model are depicted as RGB overlays. Recognizable faces have been blurred out.

smaller model with $131\times$ fewer parameters and being trained on $147\times$ fewer images. Please refer to the Supplementary Material E for qualitative results. Finally, we go beyond using MS-COCO ground-truth BB layouts and we combine objects and scenes from different images, using the BB coordinates chosen by a user to construct creative collages, see Figure 5. Once more, M&Ms showcases outstanding controllability, quality and diversity, generating impressive scenes.

## 4 RELATED WORK

**Conditional image generation.** Text-based image generation has recently shown impressive results through training on very large datasets; see *e.g.* (Ding et al., 2021; Gafni et al., 2022; Nichol et al., 2022; Ramesh et al., 2021; Reed et al., 2016; Rombach et al., 2022). Some works additionally enable image editing by providing a mask and text (Gafni et al., 2022; Nichol et al., 2022), or editing single object attributes with text (Li et al., 2019a; 2020a). Others condition on labeled bounding boxes (Casanova et al., 2020; Frolov et al., 2021; Ma et al., 2020; Sun & Wu, 2019; 2020; Sylvain et al., 2021; Zhao et al., 2019), which enables precise control of object location, but are limited when expressing object semantics, relying on object class labels or, addition-

Table 3: Zero-shot FID ($\mathbf{FID^0}$) on MS-COCO (Lin et al., 2014) at $256 \times 256$, following (Ramesh et al., 2021). *: Results with filtered dataset. Baselines: DALL-E (Ramesh et al., 2021), GLIDE (Nichol et al., 2022), Make-A-Scene (Gafni et al., 2022) and unCLIP (Ramesh et al., 2022).

| Model | # Params | # Tr. data | $\mathbf{FID^0}$ |
|---|---|---|---|
| DALL-E | 12B | 250M | $\sim 28$ |
| GLIDE | 5B | 250M | 12.2 |
| Make-A-Scene | 4B | 35M | 11.8* |
| unCLIP | 5.2B | 650M | 10.4 |
| M&Ms | 91M | 1.7M | 16.7 |

ally, attributes (Frolov et al., 2021; Ma et al., 2020), that require further data annotation. With similar approaches, some works condition on scene graphs (Ashual & Wolf, 2019; Herzig et al., 2020; Johnson et al., 2018; Li et al., 2019b), requiring less detailed scene description. Allowing for more spatial control, but requiring more detailed annotations, semantic image synthesis methods use segmentation masks to generate photorealistic scenes (Chen & Koltun, 2017; Liu et al., 2019; Park et al., 2019; Qi et al., 2018; Schönfeld et al., 2021; Tang et al., 2020b; Wang et al., 2018; 2021), but are also limited to class labels to control object appearance. Text prompts, labeled bounding boxes, scene-graphs, or segmentation masks, moreover, provide limited control about the desired scene details, see Figure 1. Among such works, (Li et al., 2021b) combines a generic scene generator

model, and iteratively pastes objects generated by class-conditional GANs in the scene. Although using local generators and discriminators, it does not allow for expressive image-based conditioning.

**Exemplar-based image generation.**    Several works condition image generation on images or image fragments. Given an input image, IC-GAN (Casanova et al., 2021) learns to generate diverse images in the neighborhood of a conditioning instance. Despite its good image quality, the objects present in the instance and their positions are not guaranteed to be preserved in the generations. PasteGAN (Li et al., 2019b) and RetrieveGAN (Tseng et al., 2020) generate scenes by using a scene-graph and object crops. At inference time, these object crops are selected as the most similar class objects from the training set in terms of scene-graph embeddings. The objects in the generated scenes are practically copy-pasted from those in the training set offering limited diversity. Shocher et al. (2020) condition the generator on a pyramid of features extracted from an image using a pre-trained classification network. At inference time, features from an object and a background can be combined from different images. Unlike our work, results shown depict compositions with a single, image centered copy-pasted object, that forms a likely combination with the background. Moreover, the generated images are limited in quality and diversity. Qi et al. (2018) combine semantic segmentation masks with crops from the training set retrieved based on their segmentation mask. In image to image translation, segmentation masks (Tang et al., 2020a) or scene graphs together with an image patch (Li et al., 2021a) are used as guidance for the final output. In contrast to the three latter works, our approach does not require any semantic labels for training or generation.

**Image editing.**  Image editing methods most often aim to preserve the input information and to apply global style changes or local fine-grained modifications to the input image. By contrast, our goal is to generate diverse and novel images inspired by the input collage. Latent manipulation methods edit images by navigating the latent spaces of generative models in directions that correspond to semantic edits (Bau et al., 2019; Cherepkov et al., 2021). This limits the diversity of the outputs and the controllability, to only those editing operations that have been discovered. To overcome this, some works leverage joint visual-textual embedding models such as CLIP (Radford et al., 2021) to drive text-based latent manipulation (Couairon et al., 2022; Crowson et al., 2022). Other works edit images by either projecting reference images into a latent space and then merging the resulting latents locally with a mask (Kim et al., 2021), or by extracting a latent of an input collage image (Chai et al., 2021). These methods only present results for human faces and other single object datasets. Other methods optimize embeddings in the latent space to later combine information from multiple images by performing affine transformations and semantic manipulations (Abdal et al., 2019; 2020; Zhu et al., 2020; Issenhuth et al., 2021). Different works learn latent vectors that correspond to disentangled parts of an image (Singh et al., 2019; Li et al., 2020b) or learn latent blobs in an unsupervised manner to decompose scenes into semantic objects (Epstein et al., 2022); these factors are then recombined at inference time to generate new images. In contrast to these works, we do not need to intervene in the latent space to generate our composed scenes. ST-GAN (Lin et al., 2018) proposes to edit images with geometrically consistent operations. Unlike ST-GAN, we do not aim at editing an image nor geometrically constrain the outputs. Photomontage and image blending techniques (Agarwala et al., 2004; Chen & Kae, 2019; Sbai et al., 2021; Xu et al., 2017; Zhang et al., 2020; Chai et al., 2021) seamlessly combine the regions of multiple images, performing high quality blending. This is different from our objective, which is to generate new content without preserving the input images as they are.

## 5    CONCLUSION

We introduced a simple yet effective way of controlling scene generation by means of image collages, which enables fine-grained control over the position and size of the objects, as well as the appearance of both background and objects in the scene. Experimental results on OpenImages showed that M&Ms is able to generate high quality images and objects, even when the collage contains randomly selected and placed objects. We showcased better controllability than IC-GAN, while maintaining the image quality and diversity. We then showed that M&Ms enables zero-shot generalization to unseen datasets by providing quantitative and qualitative results on MS-COCO, obtaining better zero-shot FID than DALL-E (Ramesh et al., 2021) despite using two orders of magnitude fewer training data and model parameters. Finally, we evaluated our model on user-generated collages with unseen object combinations and arrangements and showed impressive image generations, further emphasizing the potential to advance content generation by leveraging image collages.

## 6 ETHICS STATEMENT

Our easy to use image generation model provides a very appealing tool for content creation and manipulation. Despite the positive impacts our model can provide to content creators and researchers, there is always potential for misuse. Regardless of not being its intended use, there is the risk that malicious actors use the model to generate harmful content and as a tool for harrassment or misinformation, by using synthesized images as if they were real. Moreover, the biases present in the training data can be magnified by generative models, by further under-representing some data categories or strengthening societal stereotypes. Although we use publicly available datasets that have been annotated and human-reviewed, there is no guarantee that all images depict safe content.

To mitigate the potential misuses of the tool we recommend building collages from images that do not depict unsafe, discriminatory and harmful content. Following the DALL-E mini (Dayma et al., 2021) model's card recommendations, our model "should not be used to intentionally create or disseminate images that create hostile or alienating environments for people. This includes generating images that people would foreseeably find disturbing, distressing, or offensive; or content that propagates historical or current stereotypes."

## 7 REPRODUCIBILITY STATEMENT

We provide additional details in Supplementary Material C to allow researchers to reproduce our work, such as dataset pre-processing and setup, architecture details, training hyper-parameters and other relevant information such as compute resources needed to train the model.

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

# Controllable Image Generation via Collage Representations: Supplementary Material

We expand the material provided in the main paper to first include a statement on the limitations of our work, in Supplementary Material A. The assets used in this work are credited in Supplementary Material B. Additional experimental details are provided in Supplementary Material C, followed by extended quantitative and qualitative results in Supplementary Materials D and E, respectively. Moreover, ablation and sensitivity studies are provided in Supplementary Material F. Finally, Supplementary Material G contains experiments training our model with no labeled data.

## A    LIMITATIONS

M&Ms leverages features from a pre-trained feature extractor and, therefore, depends on it. The possibility of jointly training M&Ms together with the feature extractor is left as future work. Moreover, we have observed that when the objects are cluttered and there is significant overlap among them, the quality of the generated object can quickly degrade.

Moreover, training on a complex dataset with multiple objects and scenes is a challenging task. It is particularly difficult to generate humans and their faces, which often appear blurry from afar. Looking at other examples in the literature, diffusion models trained on ImageNet present low quality human face generation (Dhariwal & Nichol, 2021). Even the latest text-to-image model Imagen Saharia et al. (2022) makes the following statement: "*Our human evaluations found Imagen obtains significantly higher preference rates when evaluated on images that do not portray people, indicating a degradation in image fidelity*" in their limitations section.

## B    ASSETS CREDIT AND LICENSING

In Table 4 and Table 5, we provide the links and licenses to the assets used in this work. M&Ms' training scheme follows IC-GAN (Casanova et al., 2020). M&Ms builds upon the BigGAN (Brock et al., 2019) architecture, and leverages the SwAV (Caron et al., 2020) feature extractor to obtain image, scene background and object features. Moreover, we used Faiss (Johnson et al., 2019) as a fast nearest neighbor search tool to build image and object neighborhoods. Finally, we trained our model on OpenImages (v4) (Kuznetsova et al., 2020), and tested it using collages derived from OpenImages (v4) as well as MS-COCO (Lin et al., 2014) (zero-shot dataset). To calculate the zero-shot FID metric, we followed the experimental setup provided by DM-GAN (Zhu et al., 2019) that was previously used to evaluate text-to-image zero-shot generations on MS-COCO (Gafni et al., 2022; Ramesh et al., 2022; 2021).

We also credit the individual images from the OpenImages dataset (Kuznetsova et al., 2020) or obtained from the Creative Commons image search engine that were reproduced in the paper:

- "Estombar - statues in the vineyard" by muffinn is licensed under CC BY 2.0. Appears in Figures 1, 16 and 17.
- "Free" by Charles 'Andy' Lee is licensed under CC BY 2.0. Appears in Figures 1, 16 and 17.
- "old garage Rotherham" by Chris is licensed under CC BY 2.0. Appears in Figures 1, 16 and 17.
- "The Cake Cutting ceremony" by Craig Howell is licensed under CC BY 2.0. Appears in Figure 17.
- "Behind Pullman city hall, 1970s" by Robert Ashworth is licensed under CC BY 2.0. Appears in Figure 17.
- "Golden Gate National Cemetery #6" by Pargon is licensed under CC BY 2.0. Appears in Figure 18.
- "Vulture Sitting in Tree" by Hans Dekker is licensed under CC BY 2.0. Appears in Figure 2.
- "Certoza di Pavia" by Игорь, M is licensed under CC BY 2.0. Appears in Figure 2.
- "Sunset Reeuwijk II" by Frapestaartje is licensed under CC BY 2.0. Appears in Figures 3a and 17.

- "13183597093_2428d12447_o" by U.S. Army Space and Missile Defense Command (SMDC) is licensed under CC BY 2.0. Appears in Figure 2.
- "18 juillet 2011 (K5_02084)" by Manu_H is licensed under CC BY 2.0. Appears in Figures 3a, 4 and 9.
- "National Gallery - Stockholm" by Gustav Bergman is licensed under CC BY 2.0. Appears in Figure 3b.
- "The view from my hospital room " by "Marywhotravels" is licensed under CC BY 2.0. Appears in Figure 3b.
- "Alcatraz Island" by Florian Plag is licensed under CC BY 2.0. Appears in Figure 3b.
- "Conseil Régional de Haute-Normandie" by Frédéric BISSON is licensed under CC BY 2.0. Appears in Figure 3b.
- "St. Petersburg, Undated" by Nathan Hughes Hamilton is licensed under CC BY 2.0. Appears in Figure 3b.
- "IMG_3137" by Simon Davison is licensed under CC BY 2.0. Appears in Figure 18.
- "Fritz Rebotzky house - 2401 Queets Avenue, Hoquiam WA" by JOHN LLOYD is licensed under CC BY 2.0. Appears in Figure 18.
- "Liberty Star Schooner in Boston Harbor" by massmatt is licensed under CC BY 2.0. Appears in Figures 4 and 9.
- "Boston at Night" by ReneS is licensed under CC BY 2.0. Appears in Figure 9.
- "beach" by barnyz is licensed under CC BY-NC-ND 2.0. Appears in Figures 10 and 11.
- "cacti wall" by ikarusmedia is licensed under CC BY 2.0. Appears in Figures 10 and 13.
- "Cacti at Moorten Botanical Garden - Palm Springs, California" by ChrisGoldNY is licensed under CC BY-NC 2.0. Appears in Figure 10.
- "Robot" by Andy Field (Field Office) is licensed under CC BY-NC-SA 2.0. Appears in Figure 10.
- "fortune telling robot" by Paul Keller is licensed under CC BY 2.0. Appears in Figure 10.
- "Lake McDonald Lodge" by GlacierNPS is marked with Public Domain Mark 1.0. Appears in Figure 11.
- "DSC_5102" by Shane Global is licensed under CC BY 2.0. Appears in Figure 19.
- "IndieWebCamp 2011 Food" by Aaron Parecki is licensed under CC BY 2.0. Appears in Figure 19.
- "DSC09463" by Tanner Ford is licensed under CC BY 2.0. Appears in Figure 19.
- "church" by barnyz is licensed under CC BY-NC-ND 2.0. Appears in Figure 12.
- "church" by barnyz is licensed under CC BY-NC-ND 2.0. Appears in Figures 12 and 13.
- "church" by barnyz is licensed under CC BY-NC-ND 2.0. Appears in Figure 12.

Table 4: Links to the assets used in the paper.

| Asset | Link |
| --- | --- |
| IC-GAN | https://github.com/facebookresearch/ic_gan |
| BigGAN | https://github.com/ajbrock/BigGAN-PyTorch |
| SwAV | https://github.com/facebookresearch/swav |
| Faiss | https://github.com/facebookresearch/faiss |
| OpenImages (v4) | https://storage.googleapis.com/openimages/web/index.html |
| MS-COCO | https://cocodataset.org |
| DM-GAN | https://github.com/MinfengZhu/DM-GAN |

Table 5: Assets licensing information.

| Asset | License |
|---|---|
| IC-GAN | CC-BY-NC |
| BigGAN | MIT |
| SwAV | Attribution-NonCommercial 4.0 International |
| Faiss | MIT |
| OpenImages (v4) | CC BY 2.0 (images) CC BY 4.0 (annotations) |
| MS-COCO | Terms of use: https://cocodataset.org/#termsofuse |
| DM-GAN | MIT |

## C  ADDITIONAL EXPERIMENTAL DETAILS

We describe the experimental details of our model and the experimental setup, by defining the datasets in Section C.1, the architecture details in Section C.2, training details in Section C.3 and additional information such as compute resources and neighbour sets construction in Section C.4.

### C.1  DATASETS

In order to save compute and enable faster experiment iteration, we perform the ablation and sensibility studies, as well as the model selection, on a subset of OpenImages (v4) (Kuznetsova et al., 2020), that we name OpenImages500k. The best model is then trained on the full OpenImages (v4) (Kuznetsova et al., 2020) and later evaluated on MS-COCO (Lin et al., 2014) validation set. These datasets have been filtered according to the criteria below.

**OpenImages500k.**    A central crop was applied to the approximately 1.7M images from OpenImages v4. Ground-truth bounding boxes labeled as *Occluded* or *Truncated*, smaller than 2% of the crop or positioned partly outside the crop, were discarded. After filtering out these objects, we only consider images that contain at least one object, resulting in a subset with approximately 500k images, used as background scenes, with 1.7M objects overall. Among the resulting images, 99% have at most five objects.

**OpenImages.**    All images from OpenImages v4, approximately 1.7M, were cropped around one of their ground-truth objects, chosen randomly, to obtain square crops. Any object bounding box smaller than 1% of the crop or positioned partly outside it is discarded. This results in 1.5M images with a total of 5.7M objects. Most images have up to five objects ($\sim 80\%$), and the remaining 200k images are used as scene backgrounds.

**MS-COCO validation set.**    A central crop was applied to the approximately 40k images in the validation set. Ground-truth bounding boxes smaller than 2% of the crop or positioned partly outside the crop, were discarded. Images with no objects are still used as only scene backgrounds.

### C.2  ARCHITECTURE DETAILS

As already discussed in Section 2.2, we start from the BigGAN (Brock et al., 2019) generator and discriminator architecture and provide additional details on the changes we applied to handle collages as conditioning.

**Generator.**    The input collage representation $\mathbf{H}_i$, has a shape $(B, D, H, W)$, where $B$ is the batch size, $D$ the feature dimensionality – we use $D = 2048$ in our experiments –, and $H$ and $W$ are the collage height and width in pixels, respectively. We modified the normalization layers to take as input the collage representations, by first reducing the feature dimensionality of $\mathbf{H}_i$ from 2048 to 512 to limit the number of extra parameters, similarly as in IC-GAN (Casanova et al., 2021). After concatenating the noise tensor $\mathbf{Z}_i$ with the reduced-dimensionality collage representation, two $1 \times 1$ convolutional layers – instead of two fully connected layers as in BigGAN – output the gain and bias for the conditional normalization.

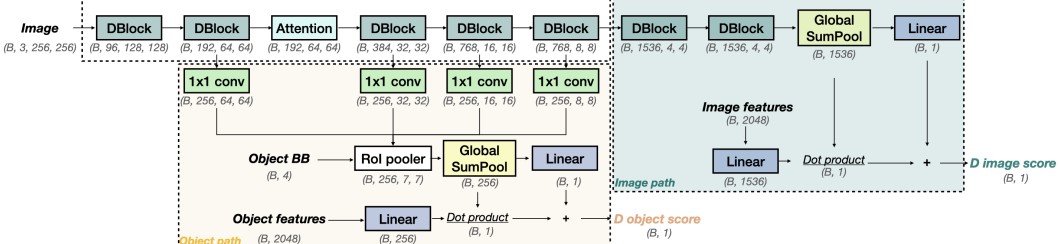

Figure 6: Image and object discriminators architecture for M&Ms at $256 \times 256$, that outputs an image score and an object score. As input, the discriminators are given an image and its features, and an object feature vector as well as its bounding box (BB). **DBlock**: block with a residual connection, where a ReLU (Nair & Hinton, 2010), a convolutional layer, a second ReLU and a second convolutional layer are applied, before optionally downsampling the feature maps. As in BigGAN (Brock et al., 2019), the convolutional (**conv**) layers and **linear** layers are spectrally normalized. **Attention**: global attention block, as used in BigGAN. **RoI pooler** (Ren et al., 2015) extracts object features, and **Global SumPool** performs a sum operation over the spatial dimensions. Different colored blocks are aimed at easing the visualization. Feature map sizes are specified below each block.

**Discriminators.** The image and object discriminators share a common path, composed of several blocks, that then branches into an image processing path and an object processing path, as depicted in Figure 6. The common path receives an image as an input, that is processed by a residual block chain. On the one hand, the image path further transforms the output of the common path and later multiplies it by the image features, fed as an input to the image path, with a dot product. These projected features are then summed with the output of a final linear layer, that processes the features without projection. On the other hand, the object path aggregates feature maps at the output of four intermediate blocks from the common path, with the help of a RoI pooler, that uses the input object bounding boxes. Finally, those aggregated features are projected, by means of a dot product, with the input object features. Similarly as for the image path, those projected features are summed with the output of a linear layer that transforms the features without projection.

## C.3 TRAINING DETAILS

We use SwAV (Caron et al., 2020), a ResNet50 trained with self-supervision, to extract features of real images as well as background scenes and foreground objects in the collage. We represent scenes and real images by means of the $1 \times 2048$ feature vector obtained after the average pooling of the ResNet50. We represent each object by means of a $1 \times 2048$ feature vector extracted by applying a RoI pooling layer (Ren et al., 2015) on the $7 \times 7 \times 2048$ feature map from the last ResNet block. The cardinalities of the nearest neighbour sets $\mathcal{A}_i^x$ and $\mathcal{A}_{i,c}^o$ are 50 and 5, respectively. We train our model by sampling up to five objects per image – as $\sim 80\%$ of the images in OpenImages contain five objects or less – and use $\lambda = 0.5$ in Equation 2. We use the default optimization hyper-parameters of IC-GAN (Casanova et al., 2021), and use early stopping on the $\text{FID}_x$ metric. We use horizontal flips as data augmentation.

## C.4 ADDITIONAL EXPERIMENTAL DETAILS

Here we specify the compute resources used to train our models and how the nearest neighbor sets were obtained.

**Compute resources.** To train M&Ms at $128 \times 128$ on OpenImages500k, we used 95h compute time on 32 Volta 32GB GPUs; to train M&Ms at $128 \times 128$ on Openimages, we used 188h compute time with 32 GPUs. Finally, to train M&Ms at $256 \times 256$ on OpenImages, we used 188h compute time with 128 GPUs.

**Features and nearest neighbor sets.** Following the practise in (Casanova et al., 2021), both background scene features $\mathbf{h}_i^s$ and object features $\mathbf{h}_i^o$, with feature dimensionality 2048, are divided

by their norm. Faiss (Johnson et al., 2019) was used to find k-NN datapoints per scene and object, that form $\mathcal{A}_i^x$ and $\mathcal{A}_{i,c}^o$ respectively, with cosine similarity as a metric.

## D    EXTENDED QUANTITATIVE RESULTS

### D.1    FID SCORES ON THE OPENIMAGES VALIDATION SET

As additional generalization results, Table 6 provides image and object FID scores for IC-GAN and M&Ms on the OpenImages validation set. M&Ms presents a slight decrease in terms of image FID compared to IC-GAN, while it greatly improves on object FID scores. These results further confirm that M&Ms enables control with respect to the baseline, achieving better object quality and diversity, while slightly trading off the quality and diversity of the global scene.

Note that unlike the numbers reported in Table 1, that samples 50K datapoints for real and generated samples, the validation set only contains 41K real samples and we compute the FID metric by sampling 40K generated and real samples. Therefore, numbers from Table 1 and the numbers in this table cannot be directly compared.

Table 6: Quantitative results on the validation set of OpenImages.

|  | $\downarrow\mathbf{FID_x}$ | $\downarrow\mathbf{FID_o}$ |
|---|---|---|
| | $128{\times}128$ | |
| IC-GAN | $\mathbf{13.3} \pm 0.1$ | $22.4 \pm 0.1$ |
| M&Ms (ours) | $14.7 \pm 0.1$ | $\mathbf{9.3} \pm 0.1$ |
| | $256{\times}256$ | |
| IC-GAN | $\mathbf{14.0} \pm 0.1$ | $26.7 \pm 0.2$ |
| M&Ms (ours) | $15.8 \pm 0.1$ | $\mathbf{15.1} \pm 0.1$ |

### D.2    FIDELITY BETWEEN INPUT COLLAGE AND GENERATED IMAGES

We additionally evaluate the fidelity between the input collages and the generated images to assess how semantically close the objects in the generated images and the images themselves are to those objects specified in the input collages and the input collage itself. In particular, object and image features are extracted with SwAV, from both the generated images and the input collages, and the cosine similarity is measured for each image and object correspondence between the input collage and generated images. Then, the similarities are averaged across images and objects separately.

Using the same evaluation setup as in Table 1 and Table 2 and sampling 10K input collages and generated images, we report the fidelity scores in Table 7. On the one hand, we observe that IC-GAN and M&Ms are on-par for most setups in terms of image fidelity, with IC-GAN presenting a higher similarity for three out of six setups. We hypothesize that in this case, higher fidelity does not mean that IC-GAN generates better scenes overall, but rather that the feature extractor lacks the ability to capture the collaged objects in the input collage. This would manifest when extracting global scene features, for both input collage and generated images, specially when the objects are small. When IC-GAN uses these global scene features to condition the model and generate samples, we already observe that the objects are not present, see Figure 9. When extracting the global scene features from the IC-GAN generated samples, those could be quite similar to the input collage features, as both ignore or do not capture the collaged objects well. We also computed the fidelity scores with the iBOT (Zhou et al., 2022) feature extractor instead of SwAV and obtained similar trends, showing that this hypothesis is not dependent on the SwAV feature extractor. On the other hand, M&Ms is superior to IC-GAN

Table 7: Fidelity score – measured with cosine similarity – for images and objects on OpenImages ($256{\times}256$) when using collages obtained by optionally altering either the object images or their bounding boxes (BB). Objects can be from the original image ($S_O$) or can be replaced by another object sampled from: the original object's neighbourhood ($S_N$), the original object's class ($S_C$), or the entire dataset of objects ($S_R$). We either keep the BB corresponding to the original object ($S_O$) or use the BB corresponding to the sampled object ($S_S$).

| | Object | BB | $\uparrow\mathbf{Fidelity_x}$ | $\uparrow\mathbf{Fidelity_o}$ |
|---|---|---|---|---|
| **IC-GAN** | $S_O$ | $S_O$ | $\mathbf{0.78} \pm 0.1$ | $0.53 \pm 0.2$ |
| | $S_N$ | $S_O$ | $\mathbf{0.77} \pm 0.1$ | $0.50 \pm 0.2$ |
| | $S_C$ | $S_O$ | $\mathbf{0.76} \pm 0.1$ | $0.45 \pm 0.1$ |
| | $S_N$ | $S_S$ | $\mathbf{0.78} \pm 0.1$ | $0.55 \pm 0.2$ |
| | $S_C$ | $S_S$ | $\mathbf{0.77} \pm 0.1$ | $0.45 \pm 0.1$ |
| | $S_R$ | $S_S$ | $\mathbf{0.77} \pm 0.1$ | $0.44 \pm 0.1$ |
| **M&Ms (ours)** | $S_O$ | $S_O$ | $\mathbf{0.78} \pm 0.1$ | $\mathbf{0.65} \pm 0.1$ |
| | $S_N$ | $S_O$ | $\mathbf{0.77} \pm 0.1$ | $\mathbf{0.59} \pm 0.1$ |
| | $S_C$ | $S_O$ | $0.73 \pm 0.1$ | $\mathbf{0.48} \pm 0.1$ |
| | $S_N$ | $S_S$ | $\mathbf{0.78} \pm 0.1$ | $\mathbf{0.64} \pm 0.1$ |
| | $S_C$ | $S_S$ | $0.76 \pm 0.1$ | $\mathbf{0.52} \pm 0.1$ |
| | $S_R$ | $S_S$ | $0.75 \pm 0.1$ | $\mathbf{0.51} \pm 0.1$ |

Table 8: Quantitative comparison between M&Ms and LostGANv2 (Sun & Wu, 2020) on the COCO-Stuff validation set at $256 \times 256$. M&Ms is trained on Openimages performs zero-shot inference, while LostGANv2 is trained on COCO-Stuff.

| Method | $\downarrow$**FID$_x$** | $\downarrow$**FID$_o$** | $\uparrow$**Fidelity$_x$** | $\uparrow$**Fidelity$_o$** |
|---|---|---|---|---|
| LostGANv2 | 37.99 | **27.86** | 0.73 | 0.57 |
| M&Ms (ours) | **34.75** | 33.72 | **0.80** | **0.69** |

in terms of object fidelity for all setups, evidencing
that M&Ms preserves the object semantics better than IC-GAN while maintaining roughly the same image semantics overall, indicating better controllability.

### D.3 COMPARISON WITH BB-TO-IMAGE METHODS

To provide a point of comparison with generative multi-object BB-to-image methods, we compare M&Ms with LostGANv2 (Sun & Wu, 2020), the BB-to-image method that obtains the lowest FID score on COCO-Stuff at $256 \times 256$.

**Experimental setup.** Note that M&Ms has been trained on OpenImages and we perform zero-shot inference on COCO-Stuff, while LostGANv2 is trained and tested on COCO-Stuff. We compute FID and fidelity scores for both scenes and objects. For M&Ms, the fidelity scores measure how well the semantics from the input collage are preserved in the output image. For LostGANv2, the fidelity scores measure how similar the generated images are to the ground-truth images, given that the model is conditioned on a bounding box layout and not directly on the input image. We have compared both methods on the validation set of COCO-Stuff, by generating one sample per input conditioning. Metrics are averaged across all validation scenes/objects. Additionally, similarly to the random collages created in Table 2 (last row), we build *random collages* with the COCO-Stuff validation set, by replacing "thing" (foreground) objects by random objects in the validation set, changing the bounding boxes and labels for LostGANv2 or the bounding boxes and object crop for M&Ms.

**Results.** In Table 8 we provide the results for this comparison. M&Ms is superior to LostGANv2 despite operating out of training data distribution, for image FID (FID$_x$) and fidelity scores. Lost-GANv2 obtains better object FID (FID$_o$), which is to be expected, given the distribution shift between the two datasets; for example, in COCO-Stuff animals appear much more often than in OpenImages, which is dominated by people. However, when using random collages (or the equivalent BB layout) as an input, both scene and object FIDs are better for M&Ms than LostGANv2. More concretely, LostGANv2 scores 58.42 image FID and 37.12 object FID, while M&Ms, obtains 51.49 and 30.25 for image and object FID respectively, showcasing the better generalization of M&Ms to unseen and challenging input conditionings. Finally, Figures 14 and 15 depict six conditionings for both LostGANv2 and M&Ms, showing the superiority of M&Ms in terms of image quality and diversity, despite not using semantic labels.

## E EXTENDED QUALITATIVE RESULTS

We provide other qualitative results on MS-COCO, for out-of-distribution collages on OpenImages, as well as results when interpolating features or noise vectors. Finally, we provide the description of the video demo provided as supplementary material to better illustrate the capabilities of M&Ms.

**Qualitative results on MS-COCO.** We present qualitative results on the zero-shot MS-COCO validation set in Figure 7. We first extract background scene features, object features and bounding box coordinates from the same image, in order to build the input collage conditioning. Then, we use these conditionings to perform inference with M&Ms, pre-trained on OpenImages. As we can observe, generated images preserve the layout specified by the object location and sizes, as well as the appearance of the objects and scenes in the collages.

| Collage | Sample #1 | Sample #2 | Sample #3 |

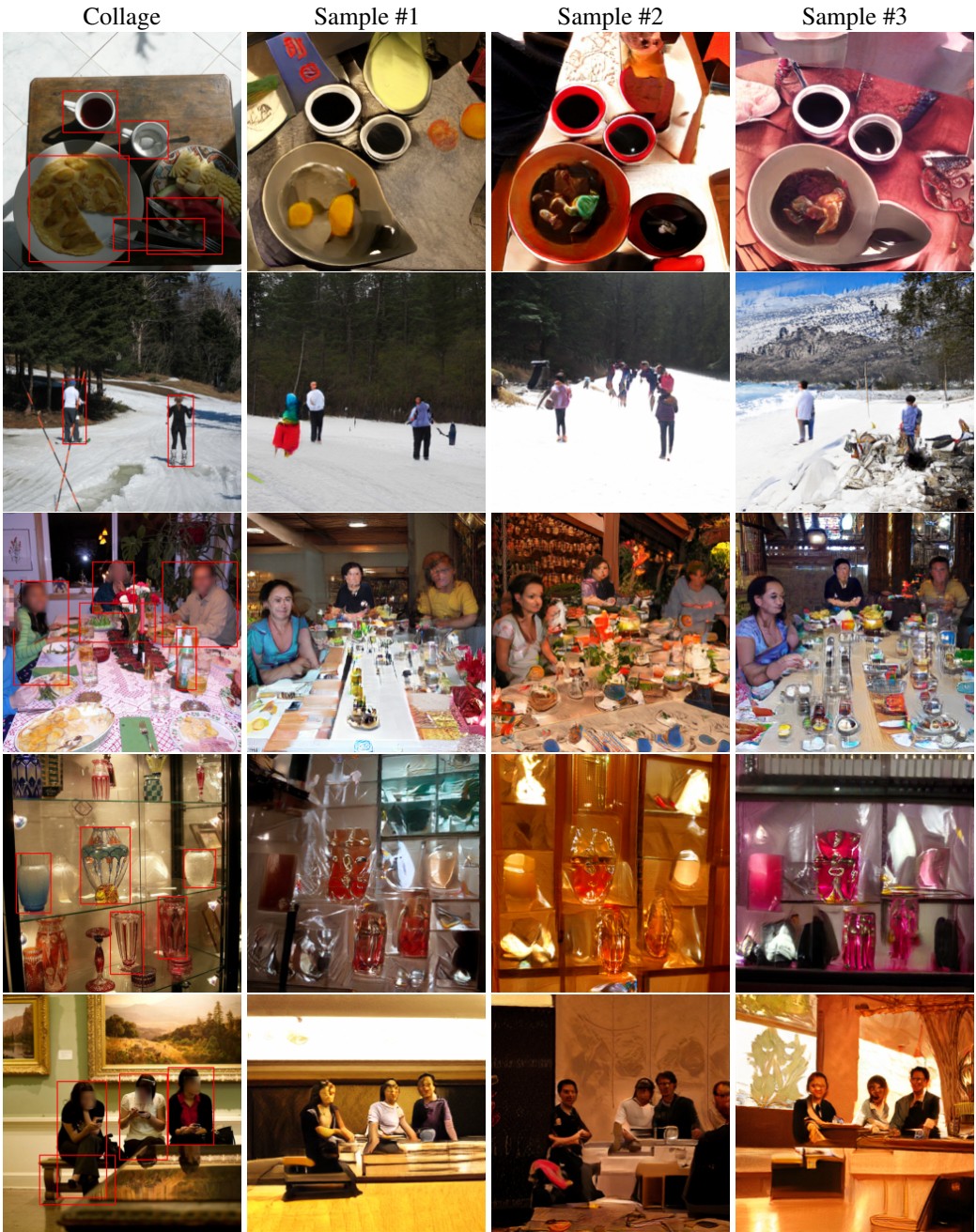

Figure 7: Five collages derived from the MS-COCO dataset and the corresponding samples from M&Ms. Note that the red boxes are overlaid on collage images to ease image interpretation. Faces in the input images have been blurred out.

We identify some cases where the quality of the generated objects is degraded, shown in Figure 8. This is the case when conditioning on objects that appear infrequently in the OpenImages (v4) dataset, such as an airplane, a skateboard, a giraffe and skis (from top to bottom). Note that these objects appear 2300, 78, 443 and 55 times, out of the total 5.7M objects used to train M&Ms. Moreover, overlapping bounding boxes result in merging objects, such as the examples shown in the second and fourth row.

| Collage | Sample #1 | Sample #2 | Sample #3 |
|---------|-----------|-----------|-----------|

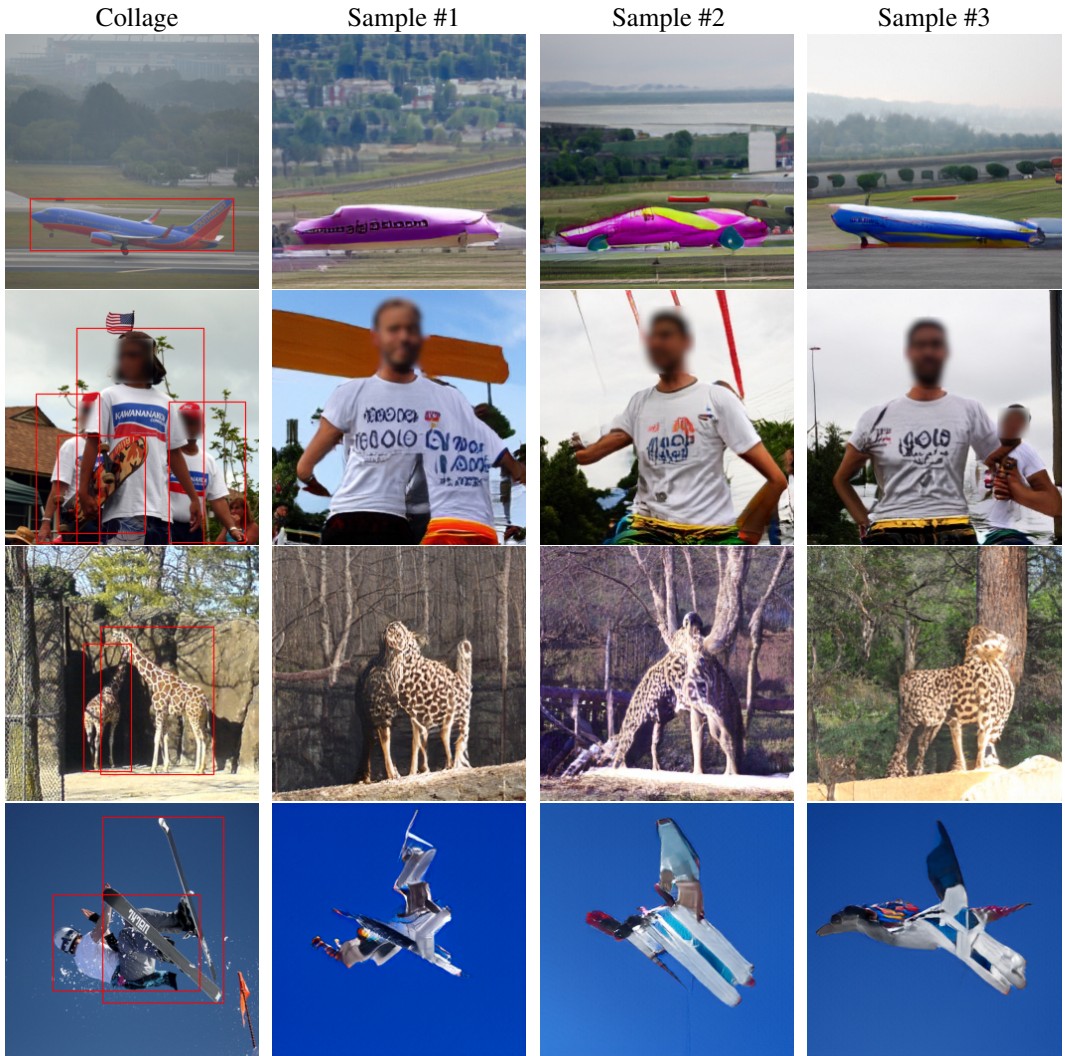

Figure 8: Four uncommon collages – with respect to OpenImages (v4) – with overlapping objects derived from the MS-COCO dataset and the corresponding samples from M&Ms. Note that the red boxes are overlaid on collage images to ease image interpretation. Faces in the input and generated images have been blurred out.

**Additional qualitative results for out-of-distribution collages on OpenImages.** To extend the qualitative results provided in Section 3.2, we provide further examples in Figure 9, where it is shown that M&Ms enables controllability even for out-of-distribution collages, unlike IC-GAN.

**Generated images using unseen classes of objects.** In Figure 10, we have used images obtained with the Creative Commons search engine to condition our model (zero-shot in the wild) and conditioned on images from two classes that are not present in OpenImages: "cactus" and "robot". Note that we do not use class labels for neither training nor inference. We observe that M&Ms is able to generate objects that preserve some characteristics of the unseen objects in the input collage, generating cacti with the same shape/textures, or generating objects that maintain the color as well as the humanoid toy aspect of the robots.

**Changing the shape of the object masks at inference time.** In Figure 11, we compare generations with the same scene background features and object features, but changing the shape and size of the object mask to non-rectangle masks. We observe that the model generalizes fairly well at inference time, adapting the shape to the one provided in the mask. Given that we never enforce the object to

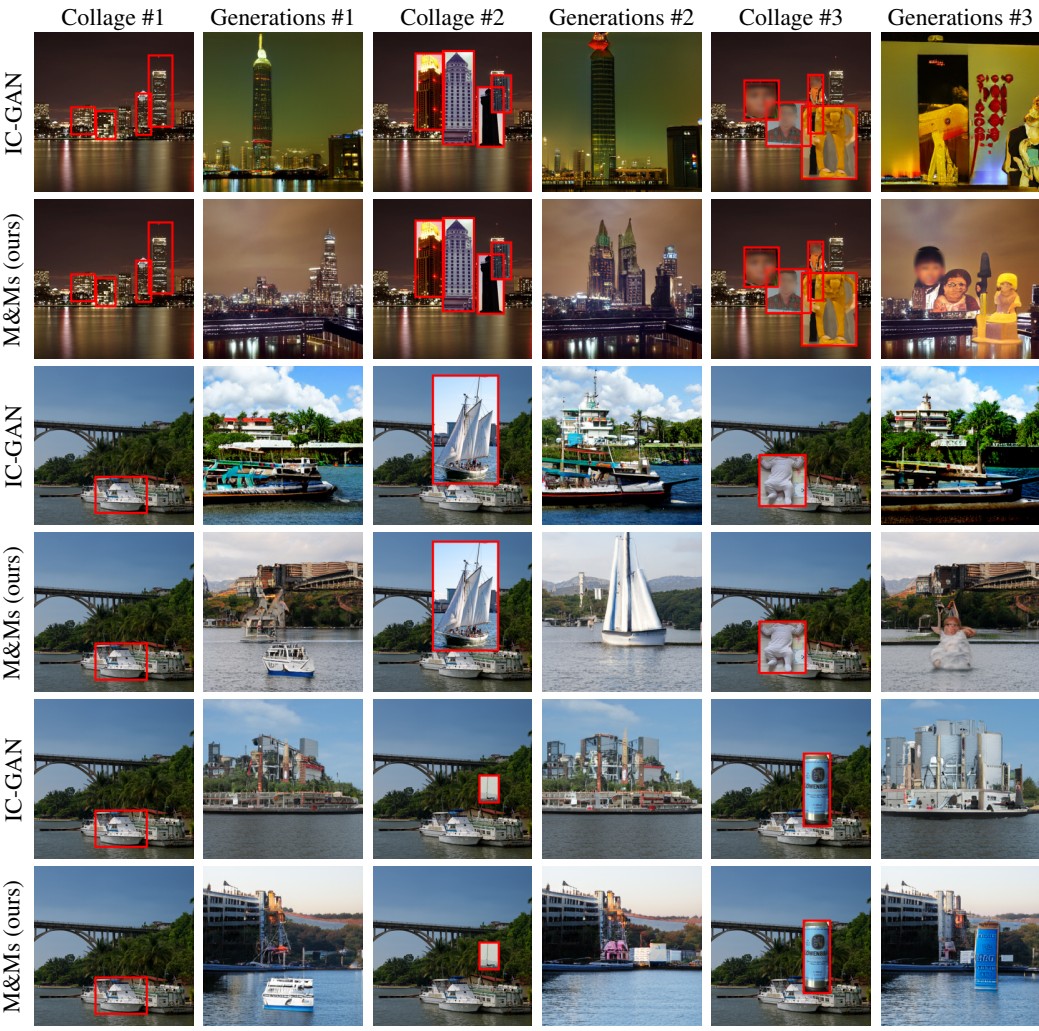

Figure 9: Example of collage conditionings and generated samples of IC-GAN and M&Ms, both trained on OpenImages (256×256). We depict three scenarios: (Collage #1) a collage with original objects; (Collage #2) a collage where the original objects have been replaced with random objects of the same class; (Collage #3) a collage where the original objects have been replaced with a random objects from the dataset. In collages #2 and #3, the bounding box of the randomly sampled object is used. Note that input collages to the M&Ms model are depicted as RGB overlays with red boxes for visualization purposes. Identifiable faces in the input and generated images have been blurred out.

| Collage | Generations | Collage | Generations |
|---|---|---|---|

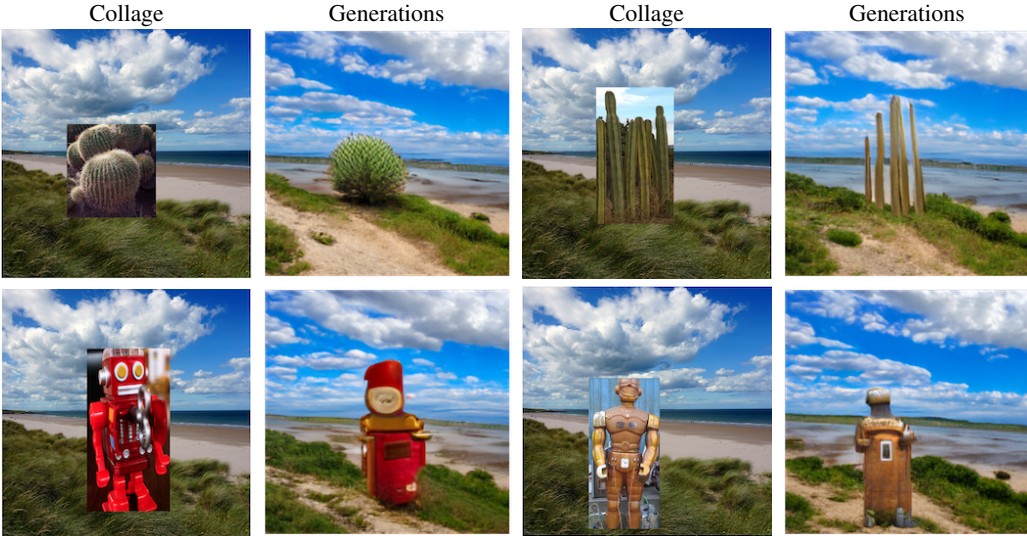

Figure 10: Collage conditionings with unseen class objects and generated samples obtained with M&Ms, trained on OpenImages (256×256). Collages are obtained by combining images collected in the wild, using the Creative Commons search engine. *Cacti* and *robot* objects, classes not present in OpenImages, are used in the first and second row respectively. Note that input collages to the M&Ms model are depicted as RGB overlays with red boxes to ease visualization.

| Collage #1 | Generation #1 | Collage #2 | Generation #2 | Collage #3 | Generation #3 |
|---|---|---|---|---|---|

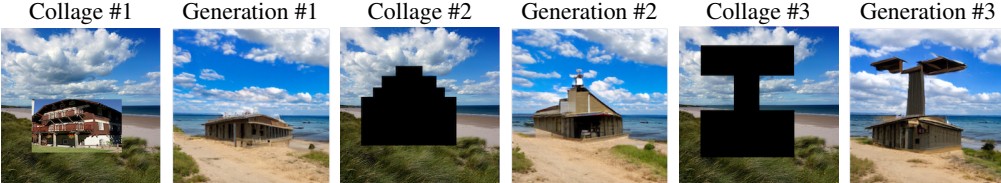

Figure 11: Collage conditionings with rectangle and non-rectangle object masks and the corresponding generated samples obtained with M&Ms, trained on OpenImages (256×256). In Collage #1, the object mask is a rectangle bounding box, illustred by copy-pasting the object in the RGB space. In Collage #2 and #3, despite using the same object features as in Collage #1, the object masks, depicted in black pixels, change shape and size. The generated object roughly adapts to the new object masks.

occupy the entirety of the input bounding box mask, the more concrete shapes specified in the input collage are approximately taken into account, but does not match them pixel by pixel.

**Feature and noise vector interpolations.** The appearance of the objects and scenes can change both by using different objects in the input collage or sampling different noise vectors. Moreover, we can obtain a smooth transition between two objects or two noise vectors by linearly interpolating them. In Figure 16, we show generated images when interpolating two noise vectors for the scene (first row), the person (second row) and the building (third row). In Figure 17, generated images are obtained by interpolating two feature vectors from collage elements: two background scenes (first row), two persons (second row) or two buildings (third row). The generated images show a smooth transition between two noise vectors or two collage element feature vectors. When those interpolations are made at the object level, the rest of the scene remains the same, enabling object morphing without altering the rest of the image.

**Comparison to image editing methods.** We compare against two image editing methods (Chai et al., 2021; Zhu et al., 2020) in Figure 12. The method in (Chai et al., 2021) can only generate one output image deterministically (first row). Moreover, the method from Zhu et al. (2020) (second row), offers little diversity between generated outputs and visually lower quality generations than M&Ms. Both image editing methods do not allow for combined changes in location, aspect ratio and size of the objects. In contrast, M&Ms generates better quality images than Zhu et al. (2020) and diverse outputs given one input collage (third row). Additionally, when changing the object size, aspect ratio

and location (fourth and fifth row), M&Ms shows increased controllability and diversity. Moreover, the editing methods in (Chai et al., 2021; Zhu et al., 2020) have been trained on single object datasets; LSUN Churches and LSUN Towers respectively (Yu et al., 2015). Therefore, they can only operate on limited image domains when performing edits, each domain needing a different trained model. As opposed to this strategy, M&Ms has been trained on a more challenging and diverse dataset, OpenImages, that contains at least 600 different objects. As shown in Figure 13, M&Ms is able to generalize to unseen classes during training, such as cacti, while both methods in Chai et al. (2021); Zhu et al. (2020) fail to successfully integrate the object in the output generations.

**Video demo.** We provide a video as part of the supplementary material to showcase a user-case example, where the user generates images with M&Ms through an iterative process. The video contains generated images for collages where objects are progressively added into a background scene and their bounding boxes are occasionally adjusted to change their shape and location. Objects and background scenes are changed throughout the video to show how these affect the generations. Moreover, noise vectors are re-sampled for each individual object and scene to control for their appearance. Note that images used as reference can either be from OpenImages or MS-COCO, and the M&Ms model has been trained on OpenImages at $256 \times 256$ resolution. Collages are displayed as an RGB overlay for simplicity.

## F  Ablation and sensitivity studies

In this section we start by performing an ablation for M&Ms, where the usage of object features is contrasted with the usage of object class labels. Then, we continue by assessing the impact of training with a single object or multiple objects per background scene. Finally, we analyze the effect of M&Ms' hyper-parameters: the cardinality of the nearest neighbor set for objects $\mathcal{A}_{i,c}^o$ and the $\lambda$ weight between the global and object losses in Equation 2. These studies are performed by training several models on the smaller OpenImages500k subset at $128 \times 128$ resolution.

**Object features versus object class labels.** We train a baseline that builds the collage representation $\mathbf{H}_i$ with background scene image features, an object bounding box and its object class label, instead of the object features as in the standard M&Ms model. Results are presented in Table 9, first and second row, where M&Ms has been trained with object features or object class labels, respectively. We observe that using object features (first row) leads to moderately worse image and object FIDs than using class labels (second row); we hypothesize that the class information is a strong conditioning that leads to a better training data fit. However, as illustrated in Figure 18, we observe little diversity between different noise samples in generated objects for M&Ms trained with object class labels. Instead, M&Ms trained with object features exhibit more diversity (different house shapes and window layouts) and furthermore, it enables fine-grained semantic control via features, evidenced by the style change of generated houses (wooden house or white paint). Overall, using object features, that have been obtained with no class label information, provides improved control and diversity compared to using class labels, that achieve moderately better FID scores, but also require costly class label annotations.

**Object set cardinality and $\lambda$.** From Table 9 we see that choosing a small object nearest neighbors set cardinality ($k^o$) results in lower object FID, and therefore we choose $k^o = 5$. The setting of $\lambda$ trades between image and object FID: with smaller $\lambda$ values the scene quality improves, while with bigger $\lambda$ values the object quality improves. Therefore, we choose an intermediate value, $\lambda = 0.5$ for all the experiments.

**Single versus multiple objects per background scene.** The collages can be built with one or more foreground collage elements (objects) per background scene. As seen in Table 10, training M&Ms with one object per background scene (first and third row) and performing inference with either one or up to five objects per background scene, M&Ms exhibits similar image and object FID. Moreover, using two, three or five objects at training time results in better image and object FID at inference time. This confirms that training with multiple objects positively impacts the quality and diversity of the generated images, even when there is only one object per background scene in the collages at inference time (second row).

Table 9: Ablation study of the usage of object (Obj.) features (F) or class labels (C), and to analyze the effect of the scene-object loss weight $\lambda$ as well as the cardinality $k^o$ of the object nearest neighbor set $\mathcal{A}_{i,c}^o$. Note that for $\lambda = 0.0$ M&Ms reverts to IC-GAN. All model variations have been trained with one object per background scene. No horizontal flips were used as data augmentation.
*: Some non-zero standard deviations are reported as 0.0 due to rounding.

| Obj. | $\lambda$ | $k^o$ | $\downarrow$**FID$_x$** | $\downarrow$**FID$_o$** |
|------|-----------|-------|-------------------------|-------------------------|
| F | 0.5 | 5 | $11.3 \pm 0.1$ | $6.6 \pm 0.0*$ |
| C | 0.5 | - | $8.9 \pm 0.1$ | $5.3 \pm 0.0*$ |
| F | 0.5 | 25 | $11.0 \pm 0.0*$ | $7.1 \pm 0.0*$ |
| F | 0.5 | 50 | $12.3 \pm 0.1$ | $7.6 \pm 0.0*$ |
| F | 0.0 | - | $8.6 \pm 0.1$ | $21.1 \pm 0.1$ |
| F | 0.1 | 5 | $9.7 \pm 0.0*$ | $7.1 \pm 0.1$ |
| F | 0.25 | 5 | $10.7 \pm 0.1$ | $7.2 \pm 0.0*$ |
| F | 0.5 | 5 | $11.3 \pm 0.1$ | $6.6 \pm 0.0*$ |
| F | 0.75 | 5 | $11.2 \pm 0.0*$ | $6.1 \pm 0.0*$ |
| F | 0.9 | 5 | $13.6 \pm 0.1*$ | $6.1 \pm 0.1$ |

Table 10: Comparison with an M&Ms trained on a single or multiple objects per background scene on OpenImages500k ($128\times128$). **O$_T$**: number of conditioning objects during training, **O$_I$**: number of conditioning objects during inference.
*: Some non-zero standard deviations are reported as 0.0 due to rounding.

| | O$_T$ | O$_I$ | $\downarrow$**FID$_x$** | $\downarrow$**FID$_o$** |
|------|-------|-------|-------------------------|-------------------------|
| M&Ms | 1 | 1 | $9.4 \pm 0.1$ | $5.3 \pm 0.0*$ |
| M&Ms | 5 | 1 | $6.3 \pm 0.0$ | $4.5 \pm 0.0$ |
| M&Ms | 1 | 5 | $9.3 \pm 0.1$ | $5.4 \pm 0.1$ |
| M&Ms | 2 | 5 | $5.6 \pm 0.0*$ | $3.6 \pm 0.0*$ |
| M&Ms | 3 | 5 | $5.6 \pm 0.0*$ | $3.7 \pm 0.0*$ |
| M&Ms | 5 | 5 | $5.6 \pm 0.0*$ | $3.9 \pm 0.0*$ |

**Different feature extractors**  We explore the usage of different feature extractors for both scene and object features. We replace the SwaV (Caron et al., 2020) feature extractor for either DINO (Caron et al., 2021) or iBOT (Zhou et al., 2022) and report the FID metrics in Table 11. Despite the improvements of DINO and iBOT regarding image classification on ImageNet with respect to SwAV, we observe no improvement over image and object FIDs when using them as feature extractors for M&Ms, and leave further exploration with additional feature extractors as future work.

**Importance of object discriminator**  We ablate the model by removing the object discriminator from M&Ms, and obtain an image FID score of $6.20 \pm 0.0$ and an object FID score of $8.11 \pm 0.1$, which are higher than M&Ms with object discriminator ($5.6 \pm 0.0$ and $3.9 \pm 0.0$ for image and object FID respectively). This confirms the importance of the object disciminator in our pipeline.

Table 11: Feature extractor ablation on OpenImages500k ($128\times128$), comparing SwAV, DINO (Caron et al., 2021) and iBOT as feature extractors for both scene and object features.
*: Some non-zero standard deviations are reported as 0.0 due to rounding.

| | **Features** | $\downarrow$**FID$_x$** | $\downarrow$**FID$_o$** |
|------|--------------|-------------------------|-------------------------|
| M&Ms | SwAV | $5.6 \pm 0.0*$ | $3.9 \pm 0.0*$ |
| M&Ms | DINO | $7.2 \pm 0.0*$ | $5.5 \pm 0.0*$ |
| M&Ms | iBOT | $6.8 \pm 0.1$ | $5.0 \pm 0.1$ |

## G   Training M&Ms without bounding box annotations

We hypothesized that M&Ms could be trained on a dataset containing images without any annotations; i.e, no ground-truth bounding box coordinates nor class labels. We perform experiments on OpenImages500K to validate the hypothesis.

### G.1   Experiments on OpenImages500K

We train three models on OpenImages500K at $128\times128$, where we replace the ground-truth bounding boxes – and the objects they define – with either: (1) random image crops, (2) bounding box coordinates predicted with Mask-RCNN (He et al., 2017) from the Detectron2 framework (Wu et al., 2019), trained on MS-COCO (Lin et al., 2014) and (3) bounding box coordinates predicted with OLN (Kim et al., 2022), trained on MS-COCO, a method that does not predict class labels and therefore it is not limited to only detecting classes in the MS-COCO training set.

We sample random crops using the ground-truth bounding box marginal distributions regarding the number of objects per image, position in the image and their size, according to the statistics of Open-Images500k. This results in roughly the same number of total objects in the dataset, approximately $900K$ objects. Bounding boxes predicted with Mask-RCNN are considered if the confidence is above 0.5 with a total of 1.4M objects, while the ones predicted with OLN are considered if the confidence is above 0.65, resulting in 2.1M objects. For the very small set of images where no prediction is found, a random crop is selected instead. In all cases, the size of the objects needs to be bigger than 2% of the image, following the standard procedure used to filter ground-truth bounding boxes in OpenImages500k. Figure 19 shows a few example images in the dataset for which bounding boxes have been extracted with the different methods.

Table 12 presents results for the three experiments, compared to M&Ms trained using ground-truth bounding boxes. At inference time, scene background features and object features are obtained from the same image, using the ground-truth object bounding boxes. This testing setup is out-of-distribution for the random, Mask-RCNN and OLN crops, and therefore the worse image and object FID metrics, compared to the model trained with ground-truth boxes, are expected. When training with random crops, the boxes are obtained by sampling from the ground-truth box distribution, which means that the bounding box distribution at inference time is in-distribution, what

Table 12: Impact study on the bounding boxes coordinates used to train M&Ms on OpenImages500k ($128\times128$). GT: ground-truth bounding boxes, RND: random crops, Mask-RCNN (He et al., 2017) predictions and OLN (Kim et al., 2022) predictions. *: Some non-zero standard deviations are reported as 0.0 due to rounding.

|       | Bounding Boxes | $\downarrow$FID$_x$ | $\downarrow$FID$_o$ |
|-------|----------------|---------------------|---------------------|
| M&Ms  | GT             | $5.6 \pm 0.0$*      | $3.9 \pm 0.0$*      |
| M&Ms  | RND            | $8.5 \pm 0.0$*      | $12.9 \pm 0.1$      |
| M&Ms  | Mask-RCNN      | $13.3 \pm 0.1$      | $8.7 \pm 0.0$*      |
| M&Ms  | OLN            | $18.0 \pm 0.1$      | $8.6 \pm 0.0$*      |

could explain the lower image FID when compared to Mask-RCNN or OLN. However, the latter methods obtain a better object FID, which could be the result of the bounding boxes being more likely to contain actual objects than the random crops baseline. Moreover, in Figure 20, we show generated images using out-of-distribution object combinations in a collage and observe that training with Mask-RCNN and OLN bounding boxes results in better object generations than when training with random crops, as already seen quantitatively in Table 12, as well as visually comparable image quality and diversity to the model trained with ground-truth bounding boxes.

Overall, these experiments showcase that M&Ms can be trained without ground-truth bounding boxes to generate reasonable scenes, see Figure 20, at the expense of worse image and object FID metrics for the in-distribution setting, compared to the model trained with ground-truth boxes.

Sample obtained with
(Chai et al., 2021)

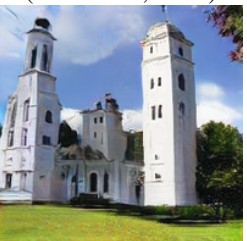

Collage      Samples obtained with in-domain image editing (Zhu et al., 2020)

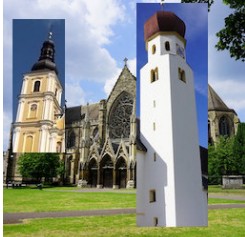 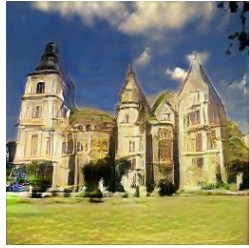 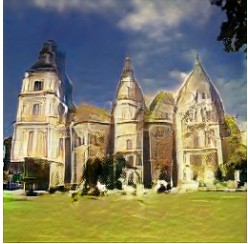 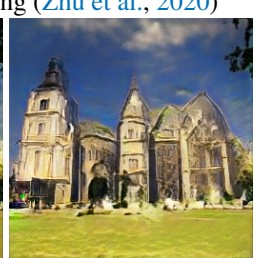

Samples obtained with M&Ms

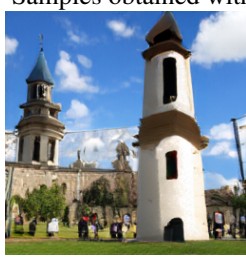 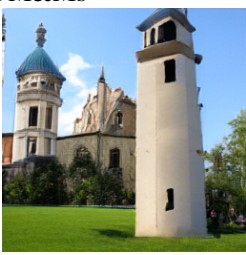 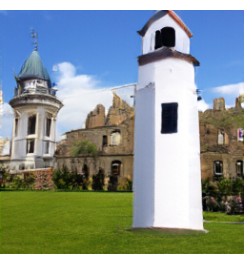

(a) Qualitative comparison with image editing methods (Chai et al., 2021; Zhu et al., 2020) that admit a collage as input. Note that (Chai et al., 2021) can only generate one deterministic output given the intput collage (first row), whereas (Zhu et al., 2020) generates images with little diversity (second row). In contrast, M&Ms offers diverse outputs given the same collage (third row). Note that neither of the image editing methods supports moving nor resizing the collage elements.

Collage    Sample w/ M&Ms    Collage    Sample w/ M&Ms

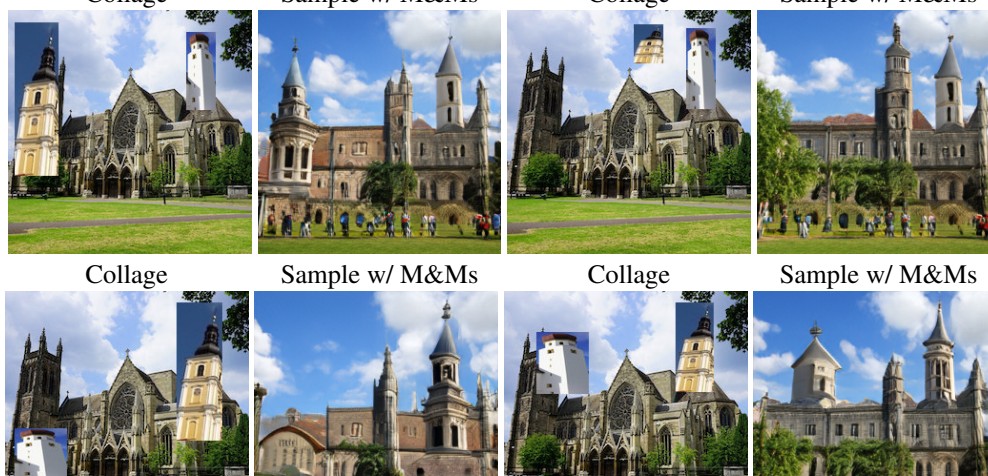

(b) M&Ms effectively handles changes in size, location and aspect ration of the collage elements.

Figure 12: The potential of M&Ms *w.r.t* image editing methods. In all cases, input collages have been build with images from the Creative Commons webpage and, for M&Ms, are depicted as RGB overlays. Discussion in Section E.

Sample obtained with
(Chai et al., 2021)

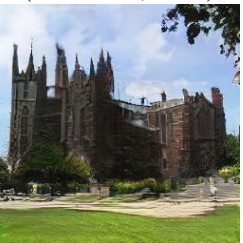

Collage          Samples obtained with in-domain image editing (Zhu et al., 2020)

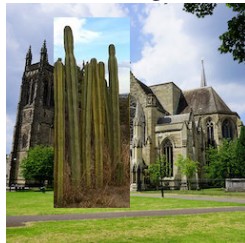 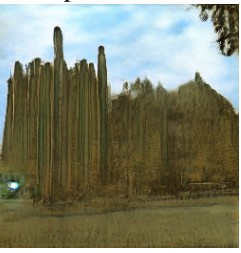 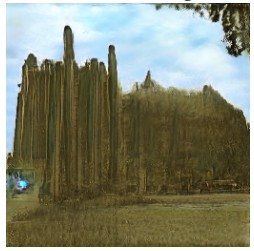 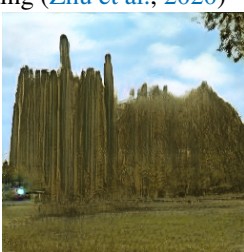

Samples obtained with M&Ms

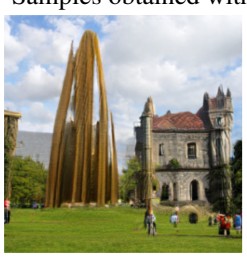 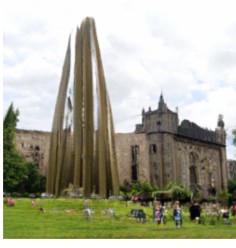 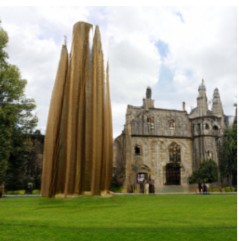

(a) Qualitative comparison with image editing methods (Chai et al., 2021; Zhu et al., 2020) that admit a collage as input. Comparisons are made with collages that include the novel class *cacti*. Note that (Chai et al., 2021) can only generate one deterministic output given the intput collage (first row), whereas (Zhu et al., 2020) generates images with little diversity (second row). Neither of the image editing methods appears to generalize to the novel object class. By contrast, M&Ms offers diverse output scenes with cacti in them (third row). Note that neither of the image editing methods supports moving nor resizing the collage elements.

Collage          Sample w/ M&Ms          Collage          Sample w/ M&Ms

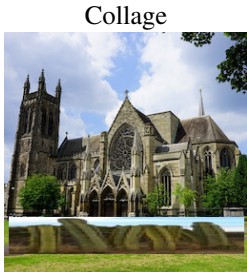 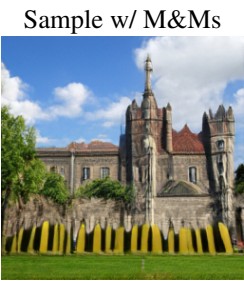 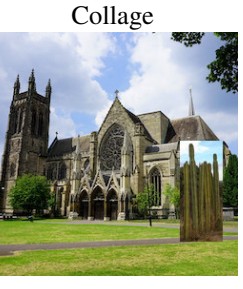 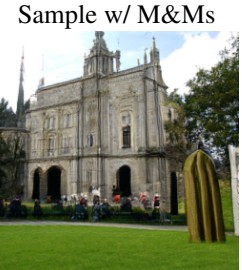

(b) M&Ms effectively handles changes in size, location and aspect ratio of the collage elements, even when those belong to novel classes which were not seen during training.

Figure 13: The potential of M&Ms *w.r.t* image editing methods when considering object classes not seen during training. Note that input collages have been build with images from the Creative Commons webpage and, for M&Ms, are depicted as RGB overlays. Discussion in Section E.

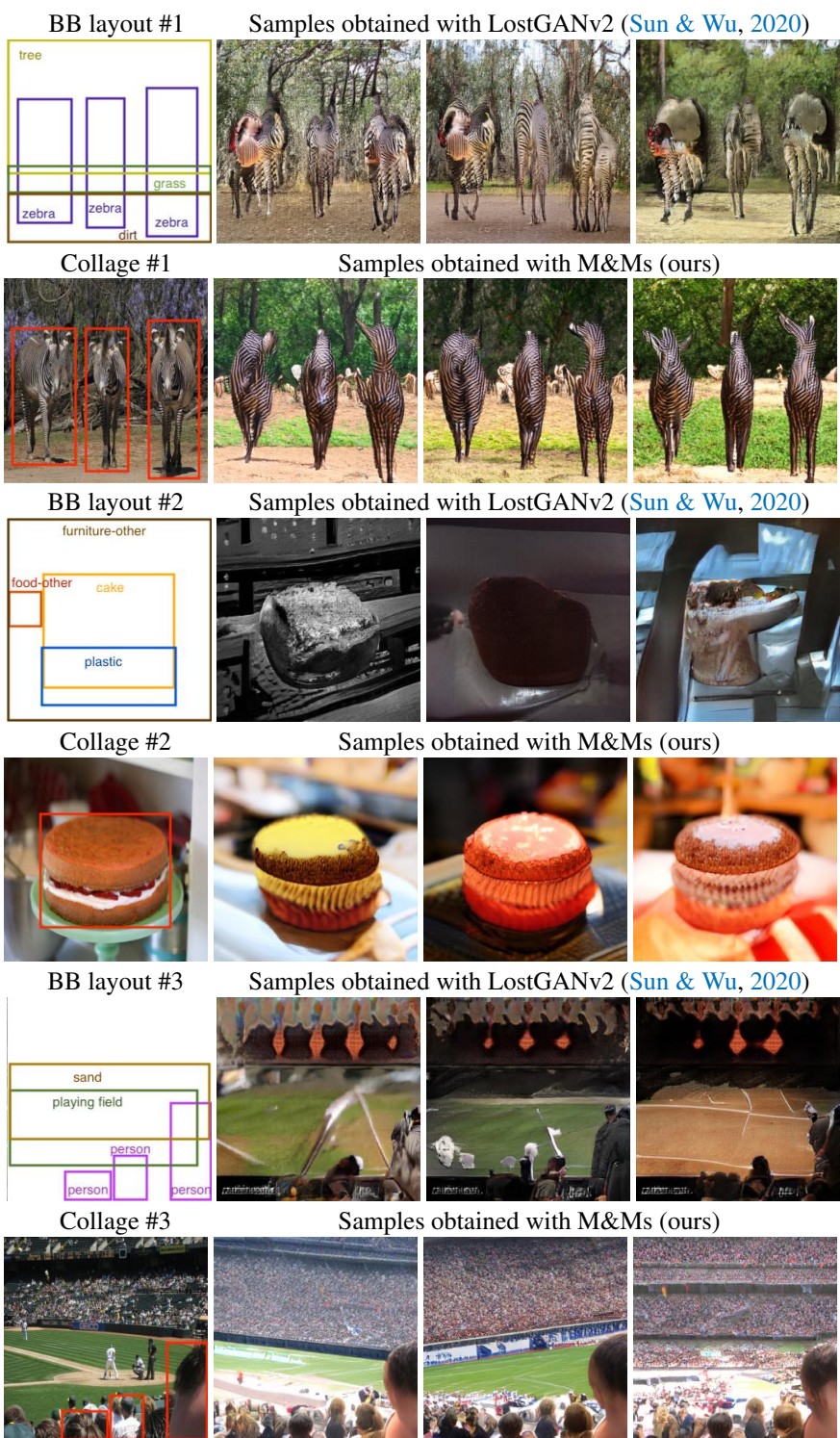

Figure 14: Qualitative results for three conditionings extracted from the COCO-Stuff validation set. As input for M&Ms, collages are composed of an image and its bounding boxes; as input for LostGANv2 (Sun & Wu, 2020), the equivalent bounding box layout is used. Note that input collages are depicted as RGB overlays. Discussion in Section D.

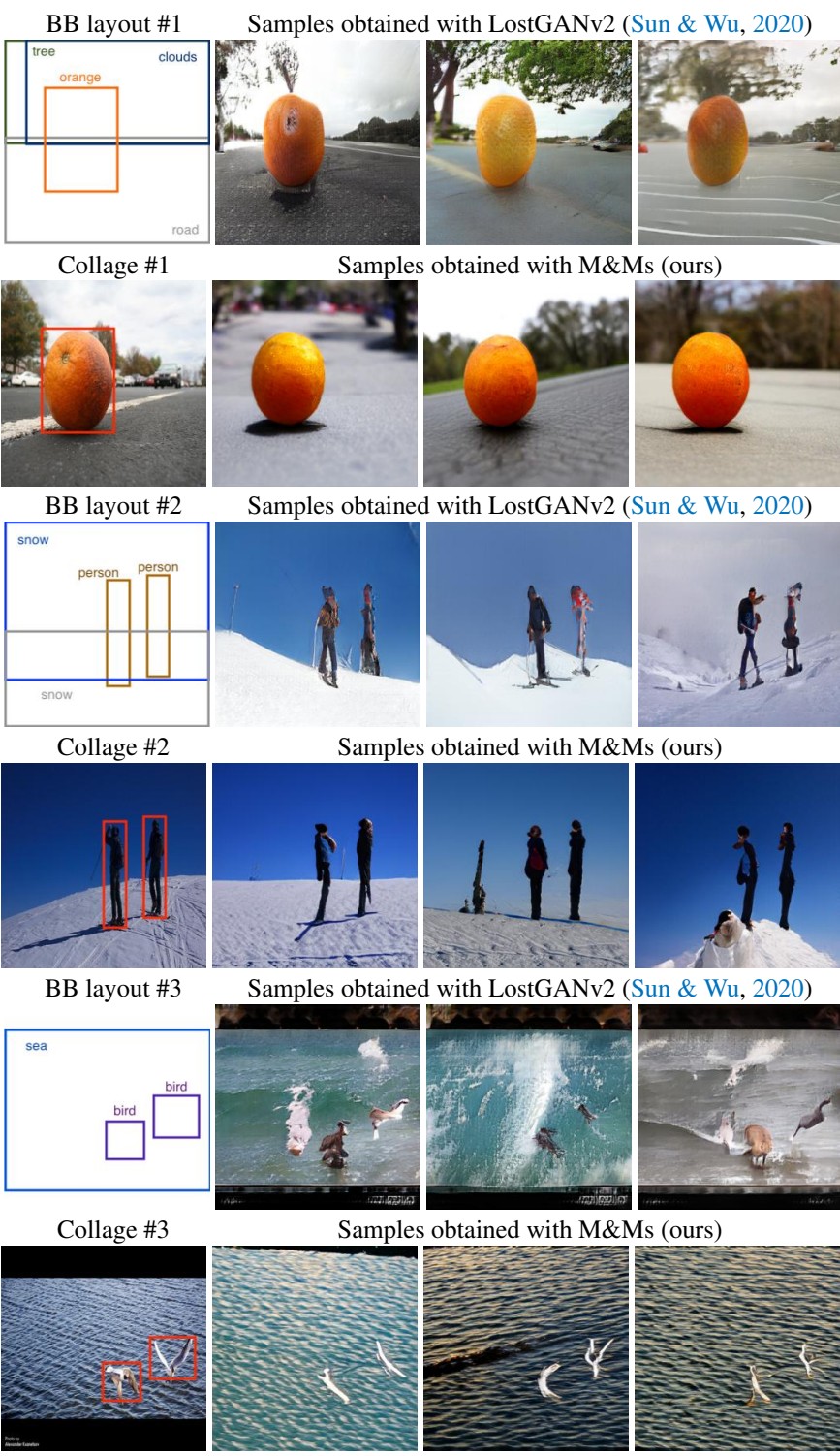

Figure 15: Extra qualitative results for three conditionings extracted from the COCO-Stuff validation set. As input for M&Ms, collages are composed of an image and its bounding boxes; as input for LostGANv2 (Sun & Wu, 2020), the equivalent bounding box layout is used. Note that input collages are depicted as RGB overlays. Discussion in Section D.

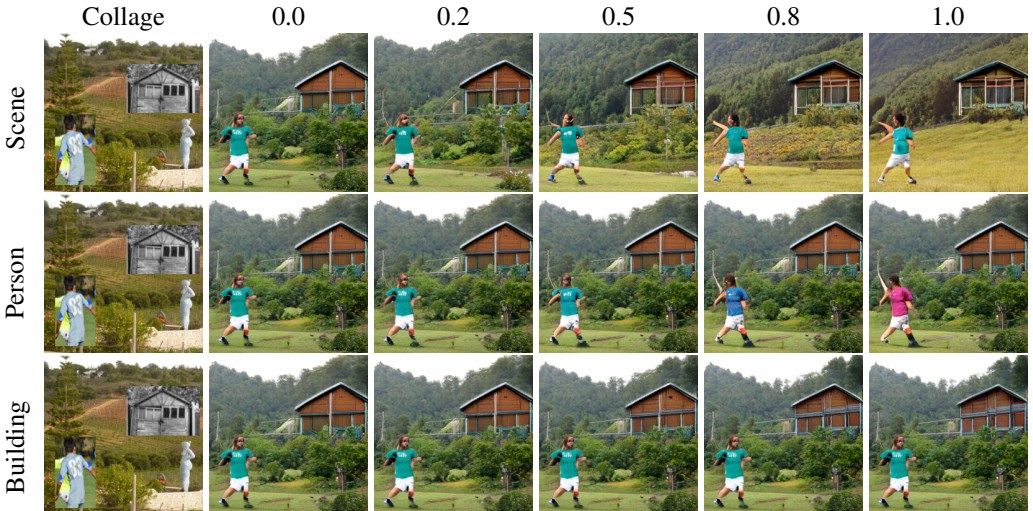

Figure 16: Noise interpolation with M&Ms conditioning on a collage built from OpenImages reference images. Generated samples are shown by interpolating noise vectors for the scene (first row), person (second row) and building (third row), with weights 0.0, 0.2, 0.5, 0.8 and 1.0 that balance between two sampled noise vectors. Discussion in Section E.

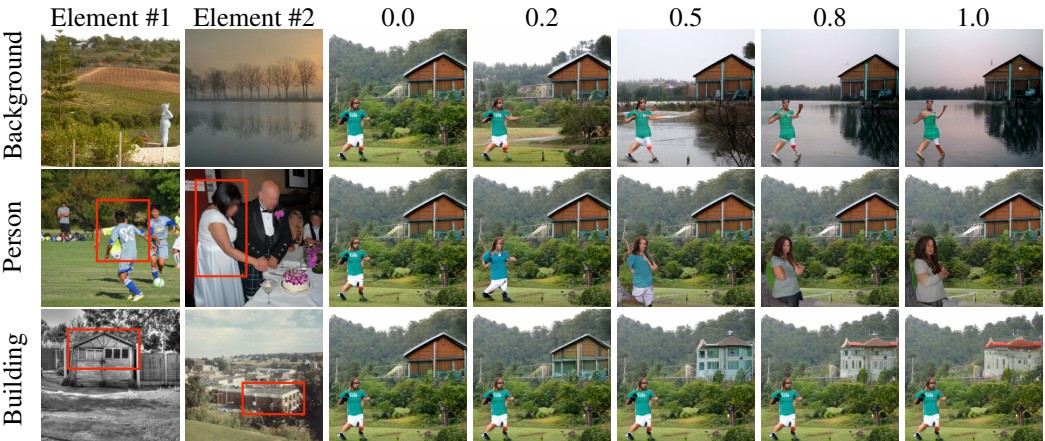

Figure 17: Feature interpolation with M&Ms conditioning on collages built from OpenImages reference images. Element #1 and #2 depict the RGB images corresponding to the features that are being interpolated: background scenes, people and buildings (first, second and third row respectively). Next to these elements, generated samples are shown by using the same noise vectors for scene and objects and performing a linear interpolation between element #1 and #2, with weights 0.0, 0.2, 0.5, 0.8, 1.0, where 0.0 means conditioning on element #1 only, and 1.0 conditioning on element #2. Faces in the input images have been blurred out. Discussion in Section E.

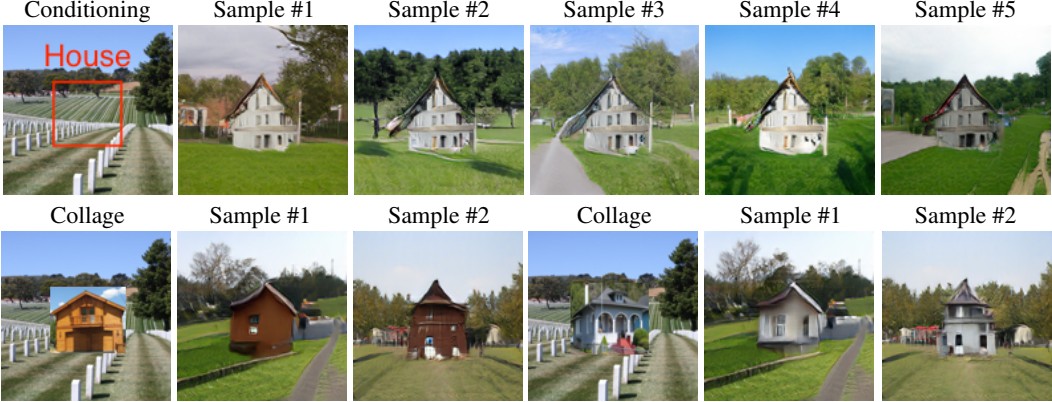

Figure 18: Collage conditionings and their respective generated samples using M&Ms trained with object class labels in the first row, and two conditionings for M&Ms trained with object features instead, in the second row. Both models have been trained on OpenImages500k (128×128). Note that input collages are depicted as RGB overlays. Discussed in Section F.

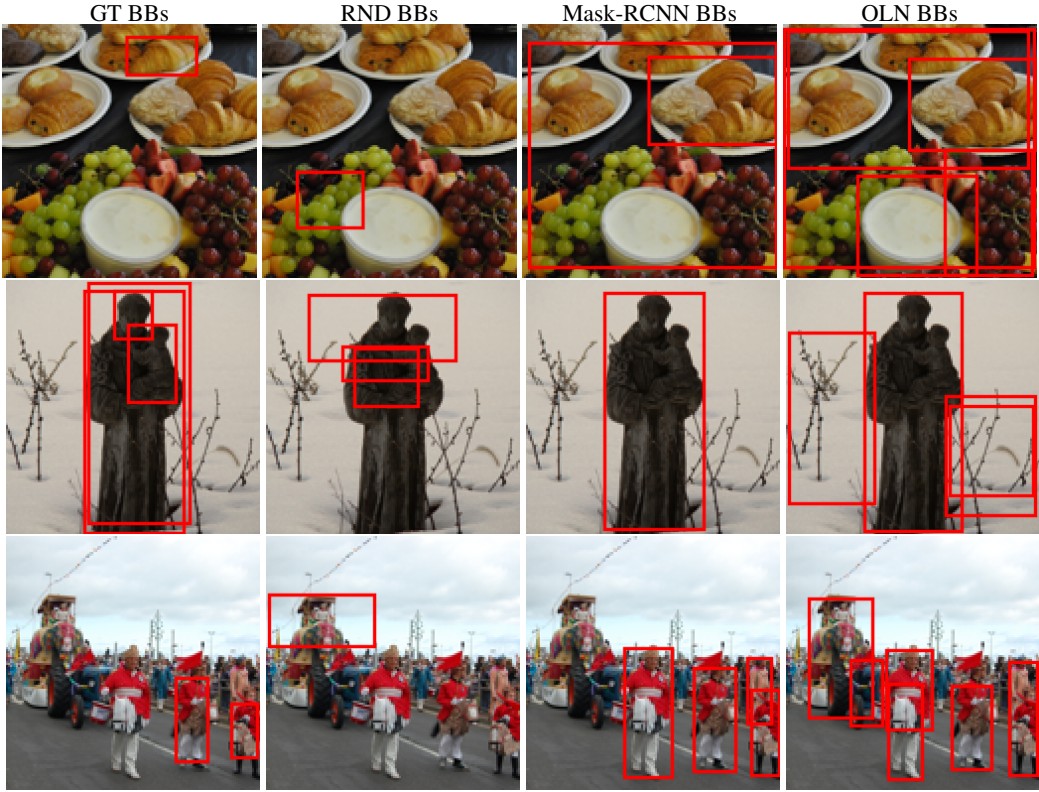

Figure 19: Images from the OpenImages500k (128×128) dataset split with their corresponding bounding boxes (BBs), overlaid as red rectangles, obtained from either the labeled ground-truth data (GT), sampled as random crops (RND), predicted by Mask-RCNN (He et al., 2017) or OLN (Kim et al., 2022). Discussion in Section G.

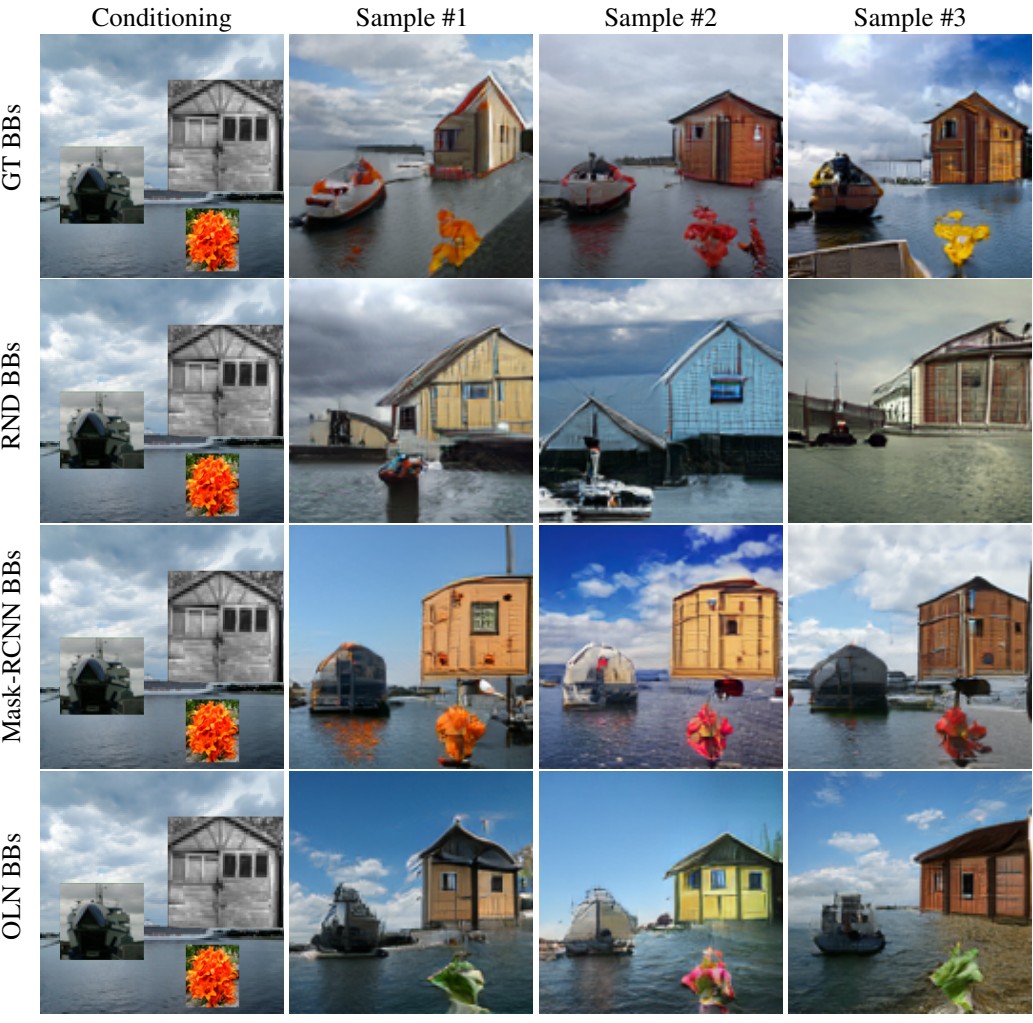

Figure 20: Generated images obtained with M&Ms trained on OpenImages ($128 \times 128$) with ground-truth bounding boxes (GT BBs), random crops (RND BBs), crops predicted by Mask-RCNN (He et al., 2017) (Mask-RCNN BBs) or predicted by OLN (Kim et al., 2022) (OLN BBs). The input conditioning has been built with ground-truth object crops from the training set in an unseen collage combination. Note that input collages are depicted as RGB overlays. Discussion in Section G.

