# OpenReview forum: "Controllable Image Generation via Collage Representations"
_ICLR.cc/2023/Conference — Submitted to ICLR 2023_

### Official Review · Reviewer_NK57 · 2022-10-25

**Confidence:** 4
**Clarity, Quality, Novelty And Reproducibility:** The paper is clear enough.
**Correctness:** 4
**Technical Novelty And Significance:** 3
**Empirical Novelty And Significance:** 3
**Recommendation:** 6

**Strength And Weaknesses:**


Overall, the idea is nice and is very practical in terms of applications in the sense that it seems more intuitive to copy/paste objects in a scene to represent the target image that you had in mind rather than describing it with a long text (that can be cumbersome and not precise enough) or drawing a segmentation mask. The idea stems from image editing, but here the goal is different as you certainly do not want the resulting image to be a smooth blending of the input collage which would not be diverse enough as a creation tool.

There are weaknesses in the experimental setup and in the overall positionning. For the experiments, the results are not extremely convincing due to the lack of comparision with methods that can do the same. Only IC-GAN is tested, but there are many more GAN inversion methods that could take the image collage as input encode it to the GAN latent space and then decode it, like:
- Lucy Chai, Jonas Wulff, and Phillip Isola. Using latent space regression to analyze and leverage compositionality in gans. arXiv:2103.10426, 2021.
- Rameen Abdal, Yipeng Qin, and Peter Wonka. Image2stylegan: How to embed images into the stylegan latent space? In Proceedings of the IEEE/CVF International Conference on Computer Vision, 2019.
- Rameen Abdal, Yipeng Qin, and Peter Wonka. Image2stylegan++: How to edit the embedded images? In Proceedings of the IEEE/CVF Conference on Computer Vision and Pattern Recognition, 2020.
- Jiapeng Zhu, Yujun Shen, Deli Zhao, and Bolei Zhou. In-domain gan inversion for real image editing. In European conference on computer vision, 2020.
- Thibaut Issenhuth, Ugo Tanielian, Jérémie Mary, David Picard. EdiBERT, a generative model for image editing. arXiv:2111.15264, 2021
- Yueqin Yin, Lianghua Huang, Yu Liu, Kaiqi Huang. DiffGAR: Model-Agnostic Restoration from Generative Artifacts Using Image-to-Image Diffusion Models. arXiv:2210.08573, 2022. (this one came out after the submission deadline, so of course it cannot be compared to)

Additionally, the experiment of fitting the training set is not convincing at all. Memorizing the training set is essentially what it takes to obtain a good score in this test and it does not tell anything about generalization capabilities. Why not just reserve a validation set and perform this metric on anything else than training data?
Overall, the visual results are not super impressive. Maybe it is because the task is very hard, but comparing with editing methods (even back to blending methods that do not use deep learning), would have shown the difficulty of trade-off between fidelity and diversity.

**Summary Of The Paper:**

This paper presents a method to generate images from a collage of patches representing objects. Each patch is fed to a feature encoder and the resulting features are spatially concatenated such that their layout corresponds that of a collage where each object occupies a specific bounding box. The aggregated feature map is then fed to a decoder that is trained adversarially such that both the full image and each bounding box corresponding to the object are realistic.

**Summary Of The Review:**

It's a good idea but the experimental validation is a bit lacking.

---
 The authors made efforts to address my concerns

---

> ### Author Response · Authors · 2022-11-15
> **Answer to reviewer NK57 [1/2]**
>
> We thank the reviewer for acknowledging and valuing the practicality of the idea and the intuitive method we propose. Moreover, we appreciate the references provided to better position our method in the literature.
>
> To address the reviewer’s comments, **changes in the manuscript** were made:
> - We have **included the references** in the related work, now reading:
> *“Image editing. Reviewed methods in this section for image editing have the goal to preserve the input information and to apply global style changes or local fine-grained modifications, as opposed to our goal: generating diverse and novel images inspired by the input collage. [...] Other works edit images by either [...], or by extracting a latent of an input collaged image (Chai et al. 2021), resulting in a final generated image, showing results for human faces and other mostly single object datasets.
> Other methods optimize embeddings in the latent space to later combine information from multiple images by perform affine transformations and semantic manipulations (Abdal et al. 2019, Abdal et al. 2020, Zhu et al. 2020, Issenhuth et al. 2021).“*
>
> - Added **direct qualitative comparison with [1] in Figures 12 and 13** (Supplementary Material). The discussion of these new results is presented in **Section E** of the Supplementary Material.
> - Added **Table 6** and discussion in Supplementary Material ** Section D.1**.
>
> We now address the concerns raised:
> > **“Lack of comparison with methods that can do the same”**
>
> - We argue these methods **cannot do the same**. First, as the reviewer already pointed out, **our goal is different** from image editing as we do not focus on maintaining the pixel-wise fidelity from the input image. Instead, **our method generates new and diverse images**, trading the fidelity required for image editing tasks for image diversity. Second, generative methods for image editing [1-6] have been trained on single object datasets, such as cars or faces, which make the task significantly easier and enable much higher quality for those objects in the restricted data domain of choice. As opposed to them, **M&Ms is trained on the challenging OpenImages** data, that contains 600 semantic categories (long-tail class distribution), arranged into complex scenes.
>
> - Having said that, we visually compare M&Ms to the method in [1] and we show the results in **Figures 12 and 13** (added to the Supplementary Material). Although both methods use collages as input, **[1] presents three important limitations when compared to ours**: 1) It is a **deterministic** method and can only generate one output image given a collage (Figure 12, first row). In contrast, **M&Ms can generate diverse** output images (Figure 12, second row).  2) Combined changes in size, aspect ratio and location of the objects are not allowed by the model in [1], while **ours is robust to those changes** and generates images that preserve those changes (Figure 12, third and fourth row). 3) In [1], each generator is **trained on a single domain dataset**, such as faces, cars or churches, and therefore needs a trained model for each considered domain. Instead, in our work we focus on a much more challenging task, training on a diverse and complex dataset such as OpenImages. As such, we are able to **generalize to classes not seen during training** (Figure 13), while the method presented in [1] fails to do so. Overall, we observe similar image quality compared to [1] in our examples in Figure 12 and better image quality in Figure 13, while addressing the aforementioned limitations of [1].
>
>
> - In **[2,4]** the focus is on image editing through latent vector manipulation. This is **orthogonal** to our proposed method and as we show in Figures 16 and 17 (Supplementary Material), we can also interpolate between latent vectors and input feature vectors for objects and scenes. Additionally, editing the semantics of the scene by discovering factors of variation in the latent space is out of scope of our work and any efforts in this direction could also benefit our method.
> - In **[3,4,5]** the focus is image editing and/or blending, as well as image inpainting, which is a **different goal** as the reviewer acknowledged.
> - In **[6]** the goal is to filter out image artefacts from input images, not generating images inspired by a collage. Therefore, it is also a **different task**.
>
> (Answer continues below)

---

> > ### Author Response · Authors · 2022-11-15
> > **Answer to reviewer NK57 [2/2]**
> >
> > > **“the experiment of fitting the training set is not convincing at all. [...] it does not tell anything about generalization capabilities. Why not just reserve a validation set and perform this metric on anything else than training data?”**
> >
> > We remark that fitting the training set (Table 1)  is only a subset of our experimental results and that **we provide results for generation to out-of-distribution input collages in Table 2 and generalization to MS-COCO in Table 3**. However, **we provide an additional table, Table 6, for the validation set of OpenImages**:
> >
> > |  Method |  FID_x |  FID_o |
> > |---|---|---|
> > | IC-GAN (128x128)  | 13.3 $\pm$ 0.1 |  22.4  $\pm$ 0.1 |
> > | M&Ms (128x128)  |  14.7 $\pm$ 0.1 |   9.3  $\pm$ 0.1 |
> > |   IC-GAN (256x256)  | 14.0 $\pm$ 0.1  | 26.7  $\pm$ 0.2  |
> > |  M&Ms (256x256) | 15.8 $\pm$ 0.1 | 15.1  $\pm$ 0.1 |
> >
> >
> > M&Ms presents a slight decrease in terms of image FID compared to IC-GAN, while it greatly improves on object FID scores. These results further confirm that M&Ms enables control with respect to the baseline, achieving better object quality and diversity, while slightly trading off the quality and diversity of the global scene.
> > Note that unlike the original Table 1, that samples 50K datapoints for real and generated samples, the validation set only contains 41K real samples and we compute the FID metric by sampling 40K generated and real samples. Therefore, numbers from Table 1 and the numbers in this table cannot be directly compared.
> >
> >
> >
> > > **“but comparing with editing methods would have shown the difficulty of trade-off between fidelity and diversity.”**
> >
> > By choosing to train on a diverse and complex dataset such as OpenImages, the **emphasis is put on generalization** to different objects and scenes, rather than training on a curated dataset of single objects, such as faces. Therefore, inherently, the **image fidelity is traded for more diversity and complexity of the output images**. Additionally, note that the impressive results of StyleGAN2 on faces do not translate to ImageNet [7,8], evidencing the importance of operating in a restricted versus diverse data domain. Moreover, editing methods provide very little (if any) diversity, and therefore are able to preserve the input image fidelity (as seen in Figures 12 and 13).
> > It is not straightforward to find a point of comparison with editing methods, given the datasets image editing models have been trained on and the one we trained M&MS on, as we have different objectives. However, we provided a point of comparison with [1] in Figures 12 and 12, as well as the discussion for the first comment of this review.
> >
> > [1] Lucy Chai, Jonas Wulff, and Phillip Isola. Using latent space regression to analyze and leverage compositionality in gans. arXiv:2103.10426, 2021.
> >
> > [2] Rameen Abdal, Yipeng Qin, and Peter Wonka. Image2stylegan: How to embed images into the stylegan latent space? In Proceedings of the IEEE/CVF International Conference on Computer Vision, 2019.
> >
> > [3] Rameen Abdal, Yipeng Qin, and Peter Wonka. Image2stylegan++: How to edit the embedded images? In Proceedings of the IEEE/CVF Conference on Computer Vision and Pattern Recognition, 2020.
> >
> > [4] Jiapeng Zhu, Yujun Shen, Deli Zhao, and Bolei Zhou. In-domain gan inversion for real image editing. In European conference on computer vision, 2020.
> >
> > [5] Thibaut Issenhuth, Ugo Tanielian, Jérémie Mary, David Picard. EdiBERT, a generative model for image editing. arXiv:2111.15264, 2021
> >
> > [6] Yueqin Yin, Lianghua Huang, Yu Liu, Kaiqi Huang. DiffGAR: Model-Agnostic Restoration from Generative Artifacts Using Image-to-Image Diffusion Models. arXiv:2210.08573, 2022.
> >
> > [7]  Casanova, Arantxa, Marlene Careil, Jakob Verbeek, Michal Drozdzal, and Adriana Romero Soriano. "Instance-conditioned gan." Advances in Neural Information Processing Systems 34 (2021): 27517-27529.
> >
> > [8] Sauer, Axel, Katja Schwarz, and Andreas Geiger. "Stylegan-xl: Scaling stylegan to large diverse datasets." In ACM SIGGRAPH 2022 Conference Proceedings, pp. 1-10. 2022.

---

> > > ### Comment · Reviewer_NK57 · 2022-11-16
> > > **Thanks for the update**
> > >
> > > Thank you for the updated version with the added results and clarification. It solves some of my concerns.
> > >
> > > I am still not really convince by the lack of comparison with editing methods (that can tackle collage). Yes, I do agree that the focus of the present work is on generalization and not fidelity whereas editing is exactly the contrary. Nevertheless they can technically take the collage as input and output a refined image. I think it would beneficial to show that with such methods, you would lack the diversity - or even the realism is case of an unusual collage - compared to the proposed approach. Something like E2EVE [a] could do the job, and if it does it with a lower quality of less diversity than the proposed approach, it's even more convincing for the proposed method. But since there is no comparison, it's just guesswork and we do not know. These editing methods may just do as well as the proposed method, that's why comparison is important. Any of [a,3,4,5] would do.
> > >
> > > [a] End-to-End Visual Editing with a Generatively Pre-Trained Artist, Andrew Brown, Cheng-Yang Fu, Omkar Parkhi, Tamara L. Berg, and Andrea Vedaldi. ECCV 2022.
> > >
> > > Side note: The appendix is difficult to parse with the figures/tables many pages away from their relevant text. It would be nice to add backward links from the figures to the corresponding text.

---

> > > > ### Author Response · Authors · 2022-11-16
> > > > **Comparison with image editing methods**
> > > >
> > > > Thank you for engaging in a dialogue with us. Please note that, as requested, we have performed comparisons with editing methods that can tackle collages. We have updated and simplified Figures 12 and 13 in our Appendix where we compared with [1] in our original rebuttal answer, and at your request we have also added additional comparisons with [4], and updated Figures 12 and 13 with those results. We have also re-written Section E to include [4] in the discussion.
> > > >
> > > > Please examine the updated Figure 12 (a), which depicts an input collage and the generated images obtained with [1]  (first row),  [4]  (second row), and our method M&Ms (third row). We observe that [1] presents no diversity, due to the fact that it is a deterministic approach, whereas [4] can generated multiple samples, but they have very little diversity. Moreover, [4], which has been trained on “LSUN Towers”, does not generalize very well to images collected from the web, despite the fact that (as the name suggests) the LSUN training set contains towers and buildings. As one can see the quality of the images generated by M&Ms is visually superior to those generated by [4]. Moreover, it is worth noting that neither of the other editing methods offer the ability to change the size, location and aspect ratio of the images. We show how M&Ms is able to handle varying sizes, locations and aspect ratios of the images in the input collage in Figure 12 (b).
> > > >
> > > > Figure 13 follows the same structure as Figure 12. Figure 13 (a) shows a qualitative comparison with image editing methods for unseen classes (cacti). In this case, we observe that [1] is unable to generalize to the unseen cactus objects (first row), and results in generations without such objects at the output image, while [4] generates images with tiled cacti all over (second row). By contrast, M&Ms is able to generalize to the unseen cactus objects (third row). The generalization to a cactus object holds even when changing its location, size or the aspect ratio drastically in Figure 13 (b), generating diverse images with cacti in their outputs.
> > > >
> > > > As an additional note, comparison with [a] is not possible, as no code for inference nor pertained models are available. In fact, the official repository says the following: “Coming soon: code for running inference on given inputs”.
> > > > Moreover, [5] has only open sourced a pretrained model trained on FFHQ (faces dataset), which is a very different data distribution than OpenImages.
> > > >
> > > > Finally, following the reviewers’ suggestion, we have added backward links in the captions of Figures 12-20 to the sections that discuss them.
> > > >
> > > > [a] End-to-End Visual Editing with a Generatively Pre-Trained Artist, Andrew Brown, Cheng-Yang Fu, Omkar Parkhi, Tamara L. Berg, and Andrea Vedaldi. ECCV 2022.
> > > >
> > > > [1] Lucy Chai, Jonas Wulff, and Phillip Isola. Using latent space regression to analyze and leverage compositionality in gans. arXiv:2103.10426, 2021.
> > > >
> > > > [4] Jiapeng Zhu, Yujun Shen, Deli Zhao, and Bolei Zhou. In-domain gan inversion for real image editing. In European conference on computer vision, 2020.
> > > >
> > > > [5] Thibaut Issenhuth, Ugo Tanielian, Jérémie Mary, David Picard. EdiBERT, a generative model for image editing. arXiv:2111.15264, 2021

---

### Official Review · Reviewer_abbj · 2022-10-27

**Confidence:** 4
**Clarity, Quality, Novelty And Reproducibility:** See above.
**Correctness:** 3
**Technical Novelty And Significance:** 3
**Empirical Novelty And Significance:** 3
**Recommendation:** 6

**Strength And Weaknesses:**

Strengths:
1. The paper is clearly written. The proposed approach is described well and looks sensible and correct. While the proposed method is an extension of the IC-GAN framework, the methodological improvement of adding object discriminators and incorporating spatial layout information leads to improvement in generation quality.
2. The model obtains significant improvement over the comparable baseline method (IC-GAN) in both in-distribution and out-of-distribution scenarios in generating objects

Comments:
1. While the model seems to obtain good FID score on the MS-COCO, I am not convinced by the comparison with text-to-image generation models like DALL-E. These are completely different models with different approaches for image generation (autoregressive, diffusion), which enable completely different image generation capability. So while its interesting to know that the FID obtained by M&Ms is better to DALL-E (for instance), it’s an apples-to-oranges comparison.
2. The examples for MS-COCO generation (in Figure 5) look qualitatively worse than those obtained by M&M on OpenImages, perhaps because these represent more cluttered scenes. Can the authors comment on how the results on MS-COCO differ from those on OpenImages?
3. How can the approach be extended to image inpainting?
4. How do you expect the method to improve with access to larger training datasets?


**Summary Of The Paper:**

This paper proposes a method for controllable generation of images conditioned on image collages as input. The proposed approach, called “mixing and matching scenes” (M&Ms), is an adversarially trained generative image model conditioned on appearance features and spatial positions of collage elements.  M&M combines these components into a coherent image, using a model based on Instance-conditioned GAN (IC-GAN, Casanov et al. 2021), but using two discriminators; a global discriminator and an object discriminator. The global discriminator encourages the generator to produce plausible and appealing images, while the object discriminator ensures  visual quality and collage-consistency of objects. In practice, the architecture follow the BigGAN generator and discriminator architectures. For experiments, M&M is trained on the 1.7 million OpenImages dataset and validated  in both in-distribution and out-of-distribution scenarios using collages derived from OpenImages and in a zero-shot dataset scenario using collages derived from MS-COCO. Performance is evaluated using image- and object-FID scores. On OpenImages, M&Ms significantly outperforms IC-GAN on object-FID, while maintaining the image-FID scores in both in-distribution and out-of-distribution scenarios. On zero-shot transfer to MS-COCO, the model also obtains low FID scores.


**Summary Of The Review:**

The paper proposes an approach for image collage-conditioned image generation using a GAN framework. Experimental results show qualitative and quantitative improvement over the baseline in generating objects in the resulting images for both in-distribution and out-of-distribution scenarios.

---

> ### Author Response · Authors · 2022-11-15
> **Answer to reviewer abbj**
>
> We profusely thank the reviewer for their feedback, and for finding the paper clear, the approach correct, and for acknowledging the improvements in both in-distribution and out-of-distribution with respect to the comparable baseline.
>
> Considering the provided feedback, **we added the following sentence** to the “zero-shot transfer to COCO” paragraph in Section 3.2: “This comparison is meant as a gauge of how M&Ms fares in this setup, rather than a direct comparison”
>
> We now address the comments in more details:
>
> > **"apples-to-oranges comparison"**
>
> We agree with the reviewer that **it’s not an apples to apples comparison**. It is challenging to compare against other methods because 1) other collage to image methods are trained on single object datasets and focus on different tasks, such as image editing and inpainting, and 2) conditional generative models trained on complex datasets have different inputs to condition the model (text, bounding boxes, masks). Our goal was to **position our method within a common benchmark for conditioning-to-image generation**, such as MS-COCO, as an extra point of reference. The Table is not meant to be taken as a direct comparison, but as a gauge of how M&Ms fares in terms of FID, number of parameters and datapoints to other existing text-to-image methods in this benchmark.
>
>
> > **"Can the authors comment on how the results on MS-COCO differ from those on OpenImages?"**
>
> It is **expected** that the quality of the generated images is worse in Figure 5 than for OpenImages, as this Figure is intended to push the boundaries of what M&Ms can do. Note that M&Ms has been trained on OpenImages, and never on MS-COCO. Therefore, the collages presented in Figure 5 as input are very out-of-distribution: 1) they are built from an **unseen dataset** during training, 2) the **objects are novel**, as they do not necessarily exist in the OpenImages training set; moreover, 3) and the **size, position and aspect ratio of the objects is changed** from the original one. Finally, 4) These objects come from different images and were collaged with the intention of presenting scenes with highly **unlikely object compositions**.
>
>
> > **image inpainting**
>
> The goal of M&Ms is to generate complex scenes with multiple objects by synthesizing diverse images inspired by the features of the input collage. Although image inpainting is an interesting task by itself, it has a different goal and therefore **it is out of the scope of this project**. In M&Ms, there is no reconstruction loss to encourage the exact same image appearance to be preserved outside of the inpainting region (also a main characteristic that differentiates our model from the image editing literature). However, if M&Ms were to be trained to
>  **fill unspecified regions of the image**, a specific **void embedding + a random mask** could be applied during training to learn how to fill out the images.
>
> > **"How do you expect the method to improve with access to larger training datasets?"**
>
> We expect the increase of data points to improve the image quality of the generated samples.

---

> > ### Author Response · Authors · 2022-11-17
> > **Follow-up with reviewer abbj**
> >
> > Dear reviewer abbj,
> >
> >  We hope our rebuttal helped resolve your comments. If there is a question or concern that remains, please let us know so we can properly address it.

---

> > > ### Comment · Reviewer_abbj · 2022-12-11
> > > **Response to rebuttal**
> > >
> > > Thank you for your detailed response to my comments -- they address the questions raised in the reviews. I have maintained my score for the paper.

---

### Official Review · Reviewer_LMNa · 2022-11-05

**Confidence:** 5
**Correctness:** 3
**Technical Novelty And Significance:** 2
**Empirical Novelty And Significance:** 2
**Recommendation:** 5

**Clarity, Quality, Novelty And Reproducibility:**

The writing is clear and easy to follow.

The overall quality is good with high-quality figures, especially and it can be improved with elaboration on equations.

The idea and the model are incremental somehow.

The model can be reproduced with the help of supplementary material.



**Strength And Weaknesses:**

Strength:

1 The model has significant control over the generation and it is clearly justified with qualitative results.

2 The generated collages are photo-realistic which is shown with particularly quantitative results.

3 Ethical concerns are addressed.

4 Limitations (pre-trained feature extraction) and failure cases (overlapping BB) are stated in the supplementary material.


Weakness:

1 The generated collages are sometimes blurry. For example, generated human is blurry in Collages 3 and 4 in Figure 2.

2 Figure 3 is misleading in that it seems that the model uses more than one object discriminator. Please redraw the figure.

3 Although the paper claims the model enables control over the position and size of the objects. It is unclear how M&Ms handles the location and the size of the objects (For example, does the model preserve the ratio of the height and width). Please elaborate on it and show more examples of object generation in different locations and with different sizes as it would be good to see how the model controls them. So, please add some results of collage generation with the same object with different locations and sizes (from small scale to big scale or maybe with the change over the ratio of height and width).

4 In Table 1, M&Ms' object FID scores are better and scene FID scores are roughly the same. Why? Does it mean that although M&Ms can generate objects with better fidelity, it is not capable to put them together? As well as, isn't it expected as IC-GAN is conditioned on only global scene features? Please add results of an ablation model which is M&Ms conditioned on only global features like IC-GAN.

5 The model of M&Ms is designed by leveraging IC-GAN, so comparison on Table 1 is like comparing with the older version M&Ms. So please include other GANs to this table.


**Summary Of The Paper:**

This paper introduces M&Ms framework that aims to generate controllable scene generation with given visual descriptions. The proposed framework takes the background scene and set of objects along with the object location and size as conditions and synthesis the photorealistic collages. In addition to realistic image generation, it is capable to control the generated collage by changing the conditions.

**Summary Of The Review:**

Although I like the idea and the model, the paper has some weaknesses related to the justification of fulling objectives (control over size and location) and the performance (Table 1). That's why my recommendation is "5: marginally below the acceptance threshold".

---

> ### Author Response · Authors · 2022-11-15
> **Answer to reviewer LMNa**
>
> We value the feedback provided by the reviewer. We appreciate the acknowledgement of the controllability and quality of generations offered by our model, shown with quantitative results.
>
> To address the reviewer’s comments, **changes in the manuscript** were made:
> - **Figure 3** was re-drawn to include “same discriminator” text in the drawing itself and additional text was added to the caption: *“Some weights are shared for the global and object discriminator (see C.2 in Supplementary Material for details).”*
> - Added **Figures 12 and 13** (as per request: more examples of object generation).
>
>
> We proceed to address the comments in detail:
>
> > **“generated human is blurry”**
>
> Training on a **complex dataset** with multiple objects and scenes is a challenging task. It is **particularly difficult to generate humans and their faces**. Looking at other examples in the literature, **diffusion models** trained on ImageNet present low quality human face generation [[1]](https://arxiv.org/pdf/2105.05233.pdf ) (4th and 5th row of their Figure 15). Even the latest text-to-image model **Imagen** [[2]](https://arxiv.org/pdf/2205.11487.pdf)  states that “*Our human evaluations found Imagen obtains significantly higher preference rates when evaluated on images that do not portray people, indicating a degradation in image fidelity*” in their limitations section.
>
> > **Unclear Figure 3**
>
> In Figure 3 it was written that the two object discriminator **share weights**, which means they are the same object discriminator, applied twice but conditioned on different object features. We considered the reviewer's suggestion and changed the text to “same discriminator”, in bold, and added this to the caption of the figure: *“Some weights are shared for the global and object discriminator (see C.2 in Supplementary Material for details).”*
>
> > **“more examples of object generation”**
>
> Location, size and aspect ratio of the objects **can be changed at will**. We have added two new figures in the Supplementary Material, **Figures 12 and 13**, that show generated images when using different sizes, location and aspect ratios for the same object in the input collage. Additionally, we would like to point out to the existing Supplementary Material provided before the rebuttal: the **shape of the mask does not need to be rectangular**, as shown in **Figure 11**, and changes in object size, location and aspect ratio are shown in the **video demo** uploaded as Supplementary Material, from 0:36 to 1:20min and 3:56 to 4:40min.
>
> > **“Table 1 discussion”**
>
>  Scene FID scores are computed with a global image representation that might not capture well the local details of the generated images, especially if those are not relevant for the underlying classification task that the Inception model used to compute the FID was trained for. Yet, the reported scene FID scores indicate that **M&Ms is able to compose objects into a meaningful scene** and with similar quality to the scenes generated by IC-GAN. At the same time, object FID scores indicate that M&Ms reproduces better the dataset object distribution when compared to the IC-GAN model – due to the enabled object control. Please note that **M&Ms only conditioned on global features corresponds to IC-GAN**, since in this case the object discriminator does not have any object features to leverage and so we only have a scene discriminator in the formulation. Therefore, we already make the comparison with the vanilla IC-GAN model in Table 1.
>
> > **“M&Ms vs IC-GAN and other GANs”**
>
>  We decided to build our collage-based model by **extending the IC-GAN formulation, which already demonstrated superiority wrt its GAN counterparts, BigGAN and StyleGAN2** on ImageNet and COCO-Stuff datasets. The comparisons **between M&Ms and IC-GAN** are meant to validate that by endowing controllability of the generation process through collage instances, **we not only improve the local (object) quality of the generations but also maintain the overall scene quality**. If the reviewer has some specific GAN method suggestions that we can use as baseline, we would greatly appreciate it.
>
> [1] Dhariwal, Prafulla, and Alexander Nichol. "Diffusion models beat gans on image synthesis." Advances in Neural Information Processing Systems 34 (2021): 8780-8794.
>
> [2] Saharia, Chitwan, William Chan, Saurabh Saxena, Lala Li, Jay Whang, Emily Denton, Seyed Kamyar Seyed Ghasemipour et al. "Photorealistic Text-to-Image Diffusion Models with Deep Language Understanding." arXiv preprint arXiv:2205.11487 (2022).
>
> [3]  Casanova, Arantxa, Marlene Careil, Jakob Verbeek, Michal Drozdzal, and Adriana Romero Soriano. "Instance-conditioned gan." Advances in Neural Information Processing Systems 34 (2021): 27517-27529.

---

> > ### Author Response · Authors · 2022-11-17
> > **Follow-up with reviewer LMNa**
> >
> > Dear reviewer LMNa,
> >
> > We hope our rebuttal helped resolve your concerns. If there is a question or concern that remains, please let us know so we can properly address it.

---

> > > ### Comment · Reviewer_LMNa · 2022-11-18
> > > **Thanks for updates**
> > >
> > > Thank you for the revised version with the adjustments and clarifications. And thank you for adding new results especially Figures 12 and 13 which convince me that M&Ms can control features of objects such as location, size, ... on the generated collages.
> > >
> > > However, I am not fully convinced of the reason of M&Ms' object FID scores are better and scene FID scores are roughly the same. Your reply leads me to think details that make objects realistic (that cause better FID scores) do not have a crucial impact on the fidelity of the whole frame. So, Is generating realistic objects unnecessary for realistic collage generation? Or are realistic details on objects lost during collage generation due to scale changes, ...?
> > >
> > > As well as I wonder why the revised manuscript does not include discussions in the reply (discussion related to “generated human is blurry” and “Table 1 discussion”).
> > >
> > > (Suggestion for comparison on Table 1) You can use GANs that you already used to qualitatively compare in Figures 12, 13, 14, and 15.
> > >
> > > (Minor) Please correct the caption of the subfigure in Figure 12, see "(b)Add text here".

---

> > > > ### Author Response · Authors · 2022-11-19
> > > > **Thanks for the follow up (part 1/2)**
> > > >
> > > > Thank you for engaging in a discussion with us, and for motivating us to take a deeper look at the results reported in Table 1.
> > > >
> > > >
> > > > **Object FID vs Image FID:**
> > > >
> > > > First of all, we would like to note that when we compute object level FIDs the images are scaled up to the input size of an inception network, so the computed FIDs will be quite sensitive to lower quality objects. When objects are composed into a scene they are smaller and, since FID is computed from the final pooling layer of an inception network, it is quite possible that higher quality objects will not have a major impact on the global FID score. Scenes generated without object position constraints also have more flexibility to compose the scene, so we don’t think it is that surprising that scene level FIDs could be similar.
> > > >
> > > > However, to zoom in into the image level results, we disentangle different aspects captured by the image FID metric by computing precision and recall [PR], for IC-GAN and M&Ms on OpenImages at 256 resolution. In this case, precision measures the percentage of generated images which lie in the estimated real data manifold, whereas recall measures the percentage of real images which lie in the estimated generated data manifold.
> > > > We obtained the following results (256x256 setup):
> > > >
> > > > |      |  Precision_x | Recall_x |
> > > > | ----------- | ----------- | ----------- |
> > > > | IC-GAN  (256)    | **69.6 +- 0.4**      | 67.6 +- 0.3|
> > > > | M&Ms (256)   | 67.3 +- 0.5        | **69.4 +- 0.4**|
> > > >
> > > > As shown in the table, M&Ms results in lower precision than IC-GAN, while exhibiting higher recall. This is perhaps unsurprising, given the M&Ms training. In particular, the M&Ms discriminator considers both objects and scene neighbours as real samples, therefore potentially pushing the overall image generations further away from the training data manifold. Intuitively, when using the same neighbourhood size at scene level, M&Ms can go beyond what is captured within this neighbourhood by introducing changes coming from the objects and their neighbours. This results in a drop in precision. At the same time, the increase in recall reflects a higher coverage as the real data lies within the generated data manifold more often.  Note that this behaviour is not captured by the FID metric which entangles both precision and recall ideas in a single divergence metric. To sum up, M&Ms results in a larger manifold of generated images, which more often goes beyond the real image manifold than the IC-GAN one.
> > > >
> > > > [PR] https://arxiv.org/pdf/1904.06991.pdf
> > > >
> > > >
> > > > **Manuscript updates:**
> > > >
> > > > - In the revised manuscript, we have included the discussion generated to “generated human is blurry” in Section A (Limitations) of the Supplementary Material.
> > > > - Regarding “Table 1 discussion”. We have updated Table 1 to include Precision and Recall numbers and changed the “fitting the training data” paragraph in Section 3.2 to account for the new discussion, that now reads:
> > > > *“In Table 1, we compare M&Ms to IC-GAN, which is conditioned on the global scene background features only. \ours achieves significantly better (lower) object FID than IC-GAN, while maintaining roughly the same image FID, highlighting the benefit of leveraging collage representations. To zoom in into the image level results, we compute precision (P_x) and recall (R_x) metrics at image level [PR]: M&Ms results in lower precision than IC-GAN, while exhibiting higher recall. This is perhaps unsurprising, given the M&Ms training. In particular, the M&Ms discriminator considers both objects and scene neighbours as real samples, therefore potentially pushing the overall image generations further away from the training data manifold. Intuitively, when using the same neighbourhood size at scene level, M&Ms can go beyond what is captured within this neighbourhood by introducing changes coming from the objects and their neighbours. This results in a drop in precision. At the same time, the increase in recall reflects a higher coverage as the real data lies within the generated data manifold more often.”*
> > > > - Due to space constraints, we've decided to remove OpenImages 128x128 results from the main body of the paper as they do not add any additional insights and just follow the trends of 256x256 (which is the image resolution we used for the rest of the results in the paper)
> > > > - We have also corrected the caption for Figure 12.

---

> > > > > ### Author Response · Authors · 2022-11-19
> > > > > **Thanks for the follow up (part 2/2)**
> > > > >
> > > > >
> > > > > **Comparisons in Table 1:**
> > > > >
> > > > > Thanks for the suggestion regarding adding more methods to Table 1. However, this would require re-training those methods for the OpenImages dataset.
> > > > > The image editing methods have been trained on very limited domain datasets (faces, churches, towers) and do not successfully generalize to unseen objects, as seen in Figure 13. Therefore, comparing against those image editing methods on OpenImages would be unfair for their method, without retraining them. Moreover, testing M&Ms in an out-of-distribution setup in their datasets would require bounding boxes, which those datasets do not have, and it would be unfair to our method (specially for face centric datasets), as we have not trained the model for those specialized cases.
> > > > >
> > > > > Moreover, training those image editing methods in a complex and diverse dataset such as OpenImages has never been done before. It is thus unclear whether such training would be even possible without major changes or significant hyper-parameter search, given how different the distribution of OpenImages and the ones of the image editing datasets is (FFHQ, LSUN Bedrooms, LSUN Churches, etc.). Therefore, we opted for a qualitative comparison on Figures 12 and 13 of the Appendix.
> > > > >
> > > > > Regarding LostGANv2, which has been trained on COCO-Stuff, it cannot be transferred out-of-distribution to OpenImages, as it requires class labels which are different from COCO. However, we do test our M&Ms in an out-of-distribution setup by comparing M&Ms and LostGANv2 on COCO-Stuff Section D.3 and Table 8 in Supplementary Material. Table 8 shows that M&Ms is superior to LostGANv2 both in terms of FID and fidelity scores, despite operating out of training data distribution. LostGANv2 obtains better object FID, which is to be expected, given the distribution shift between the two datasets; for example, in COCO-Stuff animals appear much more often than in OpenImages, which is dominated by people.  However, when using random collages (or the equivalent BB layout) as an input, both scene and object FIDs are better for M&Ms than LostGANv2. More concretely, LostGANv2 scores 58.42 image FID and 37.12 object FID, while M&Ms obtains 51.49 and 30.25 for image and object FID respectively, showcasing the better generalization capabilities of M&Ms to unseen and challenging input conditionings.

---

### Official Review · Reviewer_ovRC · 2022-11-06

**Confidence:** 2
**Correctness:** 4
**Technical Novelty And Significance:** 3
**Empirical Novelty And Significance:** 3
**Recommendation:** 5

**Clarity, Quality, Novelty And Reproducibility:**

This paper is well-organized and easy to follow. However, the novelty is limited, compared with IC-GAN.

**Strength And Weaknesses:**

# Strength:

1. This paper is well-organized and easy to follow.
2. Some quantitative experiments demonstrate the advantages of the proposed method to IC-GAN.


# Weakness:

1. The idea is incremental to IC-GAN, but the improvement is limited, especially for the qualitative evaluation.
2. The quantitative evaluation metrics are not enough, the author could provide more evaluations on the generated objects, such as classification errors.
3. More visual comparisons with other methods should be provided?


**Summary Of The Paper:**

This paper proposed a fine-grained scene image mixing generation method using visual descriptions in image collages. The M&Ms framework is optimized at the image and object level, following the generative adversarial training manner.

**Summary Of The Review:**

While this paper extends the IC-GAN to blend multiple localized distributions, the novelty is incremental, as extracting a collage representation.

---

> ### Author Response · Authors · 2022-11-15
> **Answer to reviewer ovRC [1/2]**
>
> We thank the reviewer for their feedback and appreciate they found the paper well organized and easy to follow, as well as their acknowledgment of the quantitative experiments that demonstrate the advantages of the proposed method with respect to IC-GAN.
>
> To address the reviewer’s comments, **changes in the manuscript** were made. Figures 12, 13, 14 and 15 were added to the Supplementary Material (as per the request: more visual comparisons).
>
> We now discuss each comment in detail:
>
> > **“Improvement is limited”**
>
> We respectfully disagree. The **goal** of this work is not to increase image quality but to **enable controllability at the object lev**, which is not possible with IC-GAN. We do **improve significantly on the controllability** aspect while **maintaining or surpassing IC-GAN** in terms of image quality and diversity, specially for out-of-distribution scenarios.
>
> > **“More evaluations on the generated objects such as classification errors”**
>
> Evaluating classification errors would only evaluate class correctness but not object appearance. Moreover, we point out that there is no open-sourced classification model trained for classification nor bounding box detection on OpenImages and that the class distribution is long-tailed. This means that detecting object classes correctly on real images from OpenImages **is already a challenging task** and that we cannot use a readily available classification model as a reliable probe for our model.
>
> Instead, **we evaluate object fidelity** in **Tables 7 and 8** in the Supplementary Material. Object fidelity is measured between the objects in the input collages and the objects in the generated images to assess how semantically close they are. In particular, object features are extracted with SwAV, from both the generated images and the input collages, and the cosine similarity is measured for each object correspondence between the input collage and generated images. Then, the similarities are averaged across images and objects separately.
> Using the same evaluation setup as in Table 1 and Table 2 and sampling 10K input collages and generated images, we report the fidelity scores in Table 7. On one hand, we observe that M&Ms is superior to IC-GAN in terms of object fidelity on average for all setups, with M&Ms scoring 0.65 and IC-GAN 0.55 for their best setups, respectively. This suggests that **M&Ms preserves the object semantics better than IC-GAN**, indicating better controllability. Comparisons are also made in Table 8 with LostGANv2 [1], showing **M&Ms have greater fidelity scores than LostGANv2** for objects.
>
> > **“More visual comparisons with other methods”**
>
> We add more visual comparisons in **Figures 12, 13, 14 and 15**.
> Other methods for conditional generation use other inputs  (such as bounding-boxes with class labels, text or semantic segmentation) so an apples-to-apples comparison is not possible. However, **we compare against a bounding box to image method, LostGANv2**, in Figures 14 and 15 (added to the Supplementary Material), and we discuss the comparison below.
> **We also consider image editing methods for comparison**. It is worth noting that these approaches are usually trained on single object datasets (e.g. faces, dogs, cars), which are easier to model. Their goal is to preserve the input image fidelity and focus on fine-grained manipulations, such as human face edits. In contrast, M&Ms is trained on a complex and diverse dataset, OpenImages, and as a result targets a much harder task. Moreover, the goal of M&Ms is to generate complex scenes with multiple objects by synthesizing diverse images inspired by the features of the input collage. However, we compare with [2] in Figures 12 and 13 (added to the Supplementary Material) and discuss the comparison in the following comment due to space limitations.

---

> > ### Author Response · Authors · 2022-11-15
> > **Answer to reviewer ovRC [2/2]**
> >
> > We compare visually against two additional methods:
> >
> >
> > - **LostGANv2 [1]**, a bounding box layout to image method:
> >     - In Figures 14 and 15 (Supplementary Material), six bounding box layout conditionings and their corresponding collages are used as input to LostGANv2 and M&Ms, respectively. We observe that LostGANv2 produces unrecognizable scenes in some cases, and generates poor quality images. M&Ms, however, produces diverse and good image quality outputs. We refer the reviewer to Table 7 and the last paragraph in Section D for a quantitative comparison with LostGANv2.
> >
> > - **Chai et al. [2]**, an image collage to single generated image method:
> >     - We have included a visual comparison with [2], a method suggested by  reviewer NK57, in Figures 12 and 13 (Supplementary Material). Although both methods use collages as input, [2] presents three important limitations when compared to ours: 1) It is a deterministic method and can only generate one output image given a collage (Figure 12, first row). In contrast, M&Ms can generate diverse output images (Figure 12, second row).  2) Combined changes in size, aspect ratio and location of the objects are not allowed by the model in [2], while ours is robust to those changes and generates images that preserve those changes (Figure 12, third and fourth row). 3)  In [2] each generator is trained on a single domain dataset, such as faces, cars or churches, and therefore needs a trained model for each considered domain. Instead, in our work we focus on a much more challenging task, training on a diverse and complex dataset such as OpenImages. As such, we are able to generalize to classes not seen during training (Figure 13), while the method presented in [2] fails to do so. Overall, we observe similar image quality compared to [2] in our examples in Figure 12 and better image quality in Figure 13, while addressing the aforementioned limitations of [2].
> >
> >
> >
> > Finally, if the reviewer has a specific method in mind for comparison, we would appreciate the reference. We would like to ask for a method with opensourced code that has the same goal, trained on similarly complex distributions that can also perform zero-shot transfer, similarly to our method.
> >
> > [1] Sun, Wei, and Tianfu Wu. "Learning layout and style reconfigurable gans for controllable image synthesis." IEEE transactions on pattern analysis and machine intelligence 44, no. 9 (2021): 5070-5087.
> >
> > [2] Lucy Chai, Jonas Wulff, and Phillip Isola. Using latent space regression to analyze and leverage compositionality in gans. arXiv:2103.10426, 2021.

---

> > > ### Author Response · Authors · 2022-11-16
> > > **Updated response: more visual comparisons**
> > >
> > > We would like to let the reviewer know that we have **visually compared our method with an additional image editing method [3]**.
> > >
> > > We have **updated and simplified Figures 12 and 13** in our Appendix where we compared with [1] in our original rebuttal answer, and at the request of reviewer NK57, we have also added additional comparisons with [3], and updated Figures 12 and 13 with those results. We have also re-written Section E to include [3] in the discussion.
> > >
> > > Please examine the updated Figure 12 (a), which depicts an input collage and the generated images obtained with [1] (first row), [3] (second row), and our method M&Ms (third row). We observe that [1] presents no diversity, due to the fact that it is a deterministic approach, whereas [3] can generated multiple samples, but they have very little diversity. Moreover, [3], which has been trained on “LSUN Towers”, does not generalize very well to images collected from the web, despite the fact that (as the name suggests) the LSUN training set contains towers and buildings. As one can see the quality of the images generated by M&Ms is visually superior to those generated by [3]. Moreover, it is worth noting that neither of the other editing methods offer the ability to change the size, location and aspect ratio of the images. We show how M&Ms is able to handle varying sizes, locations and aspect ratios of the images in the input collage in Figure 12 (b).
> > >
> > > Figure 13 follows the same structure as Figure 12. Figure 13 (a) shows a qualitative comparison with image editing methods for unseen classes (cacti). In this case, we observe that [1] is unable to generalize to the unseen cactus objects (first row), and results in generations without such objects at the output image, while [3] generates images with tiled cacti all over (second row). By contrast, M&Ms is able to generalize to the unseen cactus objects (third row). The generalization to a cactus object holds even when changing its location, size or the aspect ratio drastically in Figure 13 (b), generating diverse images with cacti in their outputs.
> > >
> > > [3] Jiapeng Zhu, Yujun Shen, Deli Zhao, and Bolei Zhou. In-domain gan inversion for real image editing. In European conference on computer vision, 2020.

---

> > > > ### Author Response · Authors · 2022-11-17
> > > > **Follow-up with reviewer ovRC**
> > > >
> > > > Dear reviewer ovRC,
> > > >
> > > > We hope our rebuttal helped resolve your concerns. If there is a question or concern that remains, please let us know so we can properly address it.

---

### Author Response · Authors · 2022-11-15
**Comment addressed to all reviewers**

There has been a tremendous amount of recent progress in image generation methods; yet the most visually impressive methods require significant investments in computation and in dataset size. We are **proposing a new way to control complex scene generation**, and although the comparison with prior work may not be straightforward, **we provide visual and quantitative comparisons** with the most relevant prior approaches.

By doing so, **we show the advantages in terms of controllability, generalization to unseen inputs and diversity** of the generated images with respect to **1)** a GAN conditioned on image features (IC-GAN [1]), **2)** a collage-to-image method for image editing ([2]), **3)** a bounding-box to image GAN (LostGANv2 [3]), with an **additional positioning with respect to text-to-image models in Table 3**. Moreover, we strongly believe the amount of innovation, experimentation and the manner in which we have made comparisons with prior work are **significant and of substantial interest to the ICLR community**.

We respectfully request that you **consider raising your score** if our responses have helped clear up issues that you have raised.

[1] Casanova, Arantxa, Marlene Careil, Jakob Verbeek, Michal Drozdzal, and Adriana Romero Soriano. "Instance-conditioned gan." Advances in Neural Information Processing Systems 34 (2021): 27517-27529.

[2] Lucy Chai, Jonas Wulff, and Phillip Isola. Using latent space regression to analyze and leverage compositionality in gans. arXiv:2103.10426, (2021).

[3] Sun, Wei, and Tianfu Wu. "Learning layout and style reconfigurable gans for controllable image synthesis." IEEE transactions on pattern analysis and machine intelligence 44, no. 9 (2021): 5070-5087.

---

### Decision · Program_Chairs · 2023-01-20

**Decision:**

Reject

**Justification For Why Not Higher Score:**

3 of 4 reviewers gave 5 and did not change their minds

**Justification For Why Not Lower Score:**

NA

**Metareview: Summary, Strengths And Weaknesses:**

The paper addresses fine-grained scene generation problem by providing more control at object level including its location, size and aspect ratio, hence preserves the object semantics better than the closest method IC-GAN. The authors had provided very good discussion and results in response to review comments. Though some reviewers may not be entirely satisfied with all visual and quantitative comparisons, they are still interesting. However, overall speaking, the method and results are still work-in-progress.

**Summary Of Ac-Reviewer Meeting:**

NA